# Imbalanced Adversarial Training with Reweighting

## Abstract

Adversarial training has been empirically proven to be one of the most effective and reliable defense methods against adversarial attacks. However, the majority of existing studies are focused on balanced datasets, where each class has a similar amount of training examples. Research on adversarial training with imbalanced training datasets is rather limited. As the initial effort to investigate this problem, we reveal the facts that adversarially trained models present two distinguished behaviors from naturally trained models in imbalanced datasets: (1) Compared to natural training, adversarially trained models can suffer much worse performance on under-represented classes, when the training dataset is extremely imbalanced. (2) Traditional reweighting strategies may lose efficacy to deal with the imbalance issue for adversarial training. For example, upweighting under-represented classes will drastically hurt the model's performance on well-represented classes, and as a result, finding an optimal reweighting value can be tremendously challenging. In this paper, to further understand our observations, we theoretically show that the poor data separability is one key reason causing this strong tension between under-represented and well-represented classes. Motivated by this finding, we propose Separable Reweighted Adversarial Training (SRAT) to facilitate adversarial training under imbalanced scenarios, by learning more separable features for different classes. Extensive experiments on various datasets verify the effectiveness of the proposed framework.

## 1 Introduction

The existence of adversarial samples (Szegedy et al., 2013; Goodfellow et al., 2014) has risen huge concerns on applying deep neural network (DNN) models into security-critical applications, such as autonomous driving (Chen et al., 2015) and video surveillance systems (Kurakin et al., 2016). As countermeasures against adversarial attacks, adversarial training (Madry et al., 2017; Zhang et al., 2019; Wang et al., 2019) has been empirically proven to be one of the most effective and reliable defense methods. In general, it can be formulated to minimize the model's average error on adversarially perturbed input examples (Madry et al., 2017). Although promising to improve the model's robustness, most existing adversarial training methods assume that the number of training examples from each class is equally distributed. However, datasets collected from real-world applications typically have imbalanced distribution (Everingham et al., 2010; Lin et al., 2014). Hence, it is natural to ask: *What is the behavior of adversarial training under imbalanced scenarios? Can we directly apply existing imbalanced learning strategies in natural training to tackle the imbalance issue for adversarial training?* Recent studies find that adversarial training usually presents distinct properties from natural training. For example, compared to natural training, adversarially trained models suffer more from the overfitting issue (Schmidt et al., 2018), and they tend to present strong class-wise performance disparities, even if the training examples are uniformly distributed over different classes (Xu et al., 2020a). Imagine that if the training data distribution is highly imbalanced, these properties of adversarial training can be greatly exaggerated and make it extremely difficult to be applied in practice. Therefore, it is necessary but challenging to answer aforementioned questions.

As the initial effort to study the imbalanced problem in adversarial training, in this work, we first investigate the performance of existing adversarial training under imbalanced settings. As a preliminary study shown in Section 2.1, we apply both natural training and PGD adversarial training (Madry et al., 2017) on multiple imbalanced training datasets constructed from CIFAR10 training

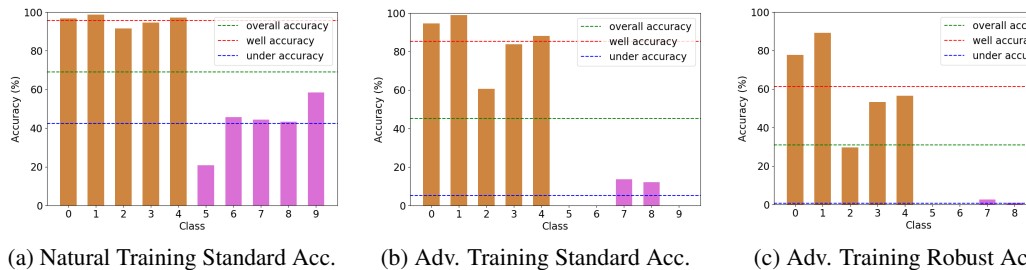

(a) Natural Training Standard Acc.     (b) Adv. Training Standard Acc.     (c) Adv. Training Robust Acc.

Figure 1: Class-wise performance of natural & adversarial training using an imbalanced CIFAR10.

dataset (Krizhevsky et al., 2009) and evaluate trained models' performance on class-balanced test dataset. From the preliminary results, we observe that, compared to naturally trained models, adversarially trained models always present very low standard & robust accuracy[1] on under-represented classes. This observation suggests that adversarial training is more sensitive to imbalanced data distribution than natural training. Thus, when applying adversarial training in practice, imbalance learning strategies should always be considered for help.

As a result, we explore potential solutions which can handle the imbalance issue for adversarial training. In this work, we focus on studying the behavior of the *reweighting* strategy (He & Ma, 2013) and leave other strategies such as resampling (Estabrooks et al., 2004) for one future work. In Section 2.2, we apply the reweighting strategy to adversarial training with varied weights assigning to one under-represented class and evaluate trained models' performance. From the results, we observe that, in adversarial training, increasing weights for an under-represented class can substantially improve the standard & robust accuracy on this class, but drastically hurt the model's performance on the well-represented class. This finding indicates that the performance of adversarially trained models is very sensitive to the reweighting manipulations and it could be very hard to figure out an eligible reweighting strategy which is optimal for all classes.

It is also worth noting that, in natural training, we find that upweighting the under-represented class increases model's standard accuracy on this class but only slightly hurts the accuracy on the well-represented class, even when adopting a large weight for the under-represent class. To further investigate the possible reasons leading to different behaviors of the reweighing strategy in natural and adversarial training, we visualize their learned features (in Figure 3), and observe that features learned by the adversarially trained model of different classes tend to mix together while they are well separated for the naturally trained model. This observation motivates us to theoretically show that when the given data distribution has poor data separability, upweighting under-represented classes will hurt the model's performance on well-represented classes. Motivated by our theoretical understanding, we propose a novel framework *Separable Reweighted Adversarial Training (SRAT)* to facilitate the reweighting strategy in imbalanced adversarial training by enhancing the separability of learned features. Through extensive experiments, we validate the effectiveness of SRAT.

## 2 PRELIMINARY STUDY

### 2.1 THE BEHAVIOR OF ADVERSARIAL TRAINING

In this subsection, we conduct preliminary studies to examine the performance of PGD adversarial training (Madry et al., 2017). Following previous works (Cui et al., 2019; Cao et al., 2019), we construct an imbalanced CIFAR10 (Krizhevsky et al., 2009) training dataset, where each of the first 5 classes (a.k.a. well-represented classes) has 5,000 training examples and each of the last 5 classes (a.k.a. under-represented classes) has 50 training examples.

Figure 1 shows the performance of naturally and adversarially trained models using a ResNet18 (He et al., 2016) architecture. From the figure, we can observe that, compared with natural training, PGD adversarial training will result in a larger performance gap between well-represented classes and under-represented classes. For example, in natural training, the ratio between the average standard

---

[1]In this work, we denote *standard accuracy* as model's accuracy on the input samples without perturbations and *robust accuracy* as model's accuracy on the input samples which are adversarially perturbed. Without clear clarification, we consider the perturbation is constrained by $l_\infty$-norm 8/255.

accuracy of well-represented classes (brown) and under-represented classes (violet) is about 2:1, while in adversarial training, this ratio expands to 16:1. Moreover, for adversarial training, it has extremely poor performance on under-represented classes. There are 3 out of the 5 under-represented classes with 0% standard & robust accuracy. As a conclusion, the performance of adversarial training is easier to be affected by imbalanced distribution than natural training and suffers more on under-represented classes. More results are reported in Appendix A.1, which further support our findings.

## 2.2 THE REWEIGHTING STRATEGY IN NATURAL TRAINING V.S. IN ADVERSARIAL TRAINING

The preliminary study in Section 2.1 demonstrates that it is highly demanding to adjust the original adversarial training methods to accommodate imbalanced data distribution. Next, we investigate the effectiveness of adopting the reweighting strategy (He & Ma, 2013) in adversarial training. Our experiments are conducted under a binary classification setting, where the training dataset contains two classes that are randomly selected from CIFAR10 dataset, with each class having 5,000 and 50 training examples respectively. Based on this training dataset, we arrange multiple trails of (reweighted) natural training and (reweighted) adversarial training, with the weight ratio between the under-represented class and well-represented class ranging from 1:1 to 200:1.

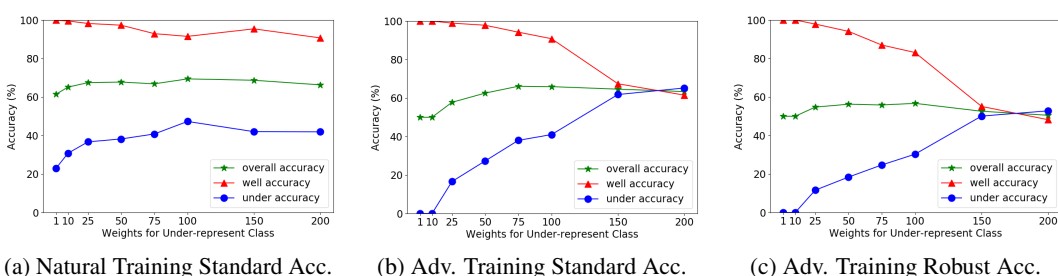

(a) Natural Training Standard Acc.     (b) Adv. Training Standard Acc.     (c) Adv. Training Robust Acc.

Figure 2: Class-wise performance of reweighted natural & adversarial training in binary classification.

Figure 2 shows the experimental results with training data sampled from the classes "cat" and "horse". As demonstrated in Figure 2, increasing the weight for the under-represented class (horse) will drastically increase the model's performance on this class, while also immensely decreasing the performance on the well-represented class (cat). For example, when increasing the weight ratio from 1:1 to 150:1, the standard accuracy of the under-represented class is improved from 0% to $\sim 60\%$ and its robust accuracy from 0% to $\sim 50\%$. However, the standard accuracy on the well-represented class drops from 100% to 60%, and its robust accuracy drops from 100% to 50%. These results illustrate that adversarial training's performance can be significantly affected by the reweighting strategy. As a result, the reweighting strategy in this setting can hardly help improve the overall performance no matter which weight ratio is chosen, because the model's performance always presents a strong tension between these two classes. More experiments using different binary imbalanced datasets are reported in Appendix A.2, where we have similar observations.

## 3 THEORETICAL ANALYSIS

In Section 2.2, we observe that in natural training, the reweighting strategy can only make a small impact on the two classes' performance. This phenomenon has been extensively studied by recent works (Byrd & Lipton, 2019; Xu et al., 2021), where they find that a linear classifier optimized by SGD on a linearly separable data will converge to the solution of the *hard-margin support vector machine* (Noble, 2006). In other words, as long as the data can be well separated, reweighting will not make huge influence on the finally trained models.

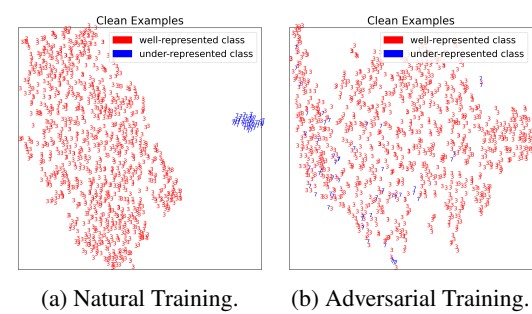

(a) Natural Training.     (b) Adversarial Training.

Figure 3: t-SNE visualization of learned features.

Inspired by their conclusions, we hypothesize that, as the adversarially trained models separate the data poorly, their performance is highly sensitive to the reweighting strategy. As a direct validation of our hypothesis, in Figure 3, we visualize the learned (penultimate layer) features of the imbalanced training examples used in the binary classification problem in Section 2.2. We find that adversarially trained models do present obviously poorer separability on the learned features. Next, we theoretically analyze the impact of reweighting on linear models which are optimized under poorly separable data and provide all detailed proof in Appendix A.3.

**Binary Classification Problem.** To construct the theoretical study, we focus on a binary classification problem, with a Gaussian mixture distribution $\mathcal{D}$ which is defined as:

$$y \sim \{-1, +1\}, \quad x \sim \begin{cases} \mathcal{N}(\mu, \sigma^2 I), & \text{if } y = +1 \\ \mathcal{N}(-\mu, \sigma^2 I), & \text{if } y = -1 \end{cases} \text{ and } \mu = (\overbrace{\eta, ..., \eta}^{\dim=d}), \quad (1)$$

where the two classes' centers ($\pm\mu \in \mathbb{R}^d$) with each dimension have mean value $\pm\eta$ ($\eta > 0$) and variance $\sigma^2$. Formally, we define the data ***separability*** as $S = \eta/\sigma^2$. Intuitively, when $S$ is larger, it suggests that two classes are well separated. Previous work (Byrd & Lipton, 2019) also closely studied this term to describe data separability.

Besides, we assume the imbalanced training dataset satisfying the condition $\Pr.(y = +1) = K \cdot \Pr.(y = -1)$ and $K > 1$, which indicates the imbalance ratio between two classes. During test, we assume two classes have the equal probability to appear. Under the data distribution $\mathcal{D}$, we will discuss the performance of linear classifiers $f(x) = \text{sign}(w^T x - b)$ where $w$ and $b$ are the weight and bias terms of the model $f$. If a reweighting strategy is involved, we define the model upweights the under-represented class "-1" by $\rho$.

**Lemma 3.1** *Under the data distribution $\mathcal{D}$ as defined in Eq. (1), with an imbalanced ratio $K$ and a reweight ratio $\rho$, the optimal classifier which minimizes the (reweighted) empirical risk:*

$$f^* = \arg\min_f \left( Pr.(f(x) \neq y | y = -1) \cdot Pr.(y = -1) \cdot \rho + Pr.(f(x) \neq y | y = +1) \cdot Pr.(y = +1) \right) \quad (2)$$

*has the solution: $w = \mathbf{1}$ and $b = \frac{1}{2}\log(\frac{\rho}{K})\frac{d\sigma^2}{\eta} = \frac{1}{2}\log(\frac{\rho}{K})\frac{d}{S}$.*

Lemma 3.1 indicates that the final optimized classifier has a weight vector equal to $\mathbf{1}$ and its bias term $b$ only depends on $K$, $\rho$ and the data separability $S$. In the following, we first focus on one special setting when $\rho = 1$, which is the original ERM model without reweighting. Specifically, we aim to compare the behavior of linear models when they can poorly separate data (like adversarial trained models) or they can well separate data (like naturally trained models).

**Theorem 3.1** *Under two data distributions $(x^{(1)}, y^{(1)}) \in \mathcal{D}_1$ and $(x^{(2)}, y^{(2)}) \in \mathcal{D}_2$ with different separabilities $S_1 > S_2$, let $f_1^*$ and $f_2^*$ be the optimal non-reweighted classifiers ($\rho = 1$) under $\mathcal{D}_1$ and $\mathcal{D}_2$, respectively. Given the imbalance ratio $K$ is large enough, we have:*

$$\begin{aligned} &Pr.(f_1^*(x^{(1)}) \neq y^{(1)} | y^{(1)} = -1) - Pr.(f_1^*(x^{(1)}) \neq y^{(1)} | y^{(1)} = +1) \\ &< Pr.(f_2^*(x^{(2)}) \neq y^{(2)} | y^{(2)} = -1) - Pr.(f_2^*(x^{(2)}) \neq y^{(2)} | y^{(2)} = +1). \end{aligned} \quad (3)$$

Intuitively, Theorem 3.1 suggests that when the data separability $S$ is low (such as $\mathcal{D}_2$), the optimized classifier (without reweighting) can intrinsically have a larger error difference between the under-represented class "-1" and the well-represented class "+1". Similar to the observation in Section 2.1 and Figure 3, adversarially trained models present a weak ability to separate data, and they also present a strong performance gap between the well-represented class and under-represented class. Conclusively, Theorem 3.1 indicates that the poor ability to separate the training data can be one important reason which leads to the strong performance gap of adversarially trained models.

Next, we consider the case when the reweighting strategy is applied. In particular, we compare the impact of upweighting the under-represented class on the performance of well-represented class.

**Theorem 3.2** *Under two data distributions $(x^{(1)}, y^{(1)}) \in \mathcal{D}_1$ and $(x^{(2)}, y^{(2)}) \in \mathcal{D}_2$ with different separabilities $S_1 > S_2$, let $f_1^*$ and $f_2^*$ be the optimal non-reweighted classifiers ($\rho = 1$) under $\mathcal{D}_1$*

and $\mathcal{D}_2$, respectively, and let $f_1'^{*}$ and $f_2'^{*}$ be the optimal reweighted classifiers under $\mathcal{D}_1$ and $\mathcal{D}_2$ given the optimal reweighting ratio ($\rho = K$). Given the imbalance ratio $K$ is large enough, we have:

$$
\begin{aligned}
&Pr.(f_1'^{*}(x^{(1)}) \neq y^{(1)}|y^{(1)} = +1) - Pr.(f_1^{*}(x^{(1)}) \neq y^{(1)}|y^{(1)} = +1) \\
&< Pr.(f_2'^{*}(x^{(2)}) \neq y^{(2)}|y^{(2)} = +1) - Pr.(f_2^{*}(x^{(2)}) \neq y^{(2)}|y^{(2)} = +1).
\end{aligned}
\tag{4}
$$

As Theorem 3.2 shows, when the data distribution has poorer data separability (such as $\mathcal{D}_2$), upweighting the under-represented class can cause greater hurt on the performance of the well-represented class. It is also consistent with our empirical findings about adversarial training models. Since the adversarially trained models poorly separate the data (Figure 3), upweighting the under-represented class always drastically decreases the performance of the well-represented class (Section 2.2). Through the discussions in both Theorem 3.1 and Theorem 3.2, we conclude that the poor separability can be one important reason which makes adversarial training and its reweighted variants extremely difficult to achieve good performance under imbalance data distribution. Therefore, in the next section, we will explore potential solutions which can facilitate the reweighting strategy in adversarial training.

## 4 SEPARABLE REWEIGHTED ADVERSARIAL TRAINING (SRAT)

The observations from both preliminary study and theoretical understandings indicate that more separable data will advance the reweighting strategy in adversarial training under imbalanced scenarios. Thus, in this section, we present a framework, Separable Reweighted Adversarial Training (SRAT), which enables the effectiveness of the reweighting strategy in adversarial training under imbalanced scenarios by increasing the separability in the learned latent feature space.

### 4.1 REWEIGHTED ADVERSARIAL TRAINING

Given an input example $(x, y)$, adversarial training (Madry et al., 2017) aims to obtain a robust model $f_\theta$ that can make the same prediction $y$ for an adversarial example $x'$, generated by applying an adversarially perturbation on $x$. The adversarial perturbations are typically bounded by a small value $\epsilon$ under $L_p$-norm, i.e., $\|x' - x\|_p \leq \epsilon$.

As indicated in Section 2.1, adversarial training cannot be applied in imbalanced scenarios directly, as it presents very low performance on under-represented classes. To tackle this problem, a natural idea is to integrate existing imbalanced learning strategies proposed in natural training, such as reweighting, into adversarial training to improve the trained model's performance on those under-represented classes. Hence, the reweighted adversarial training can be defined as

$$
\min_{\theta} \frac{1}{n} \sum_{i=1}^{n} \max_{\|x_i' - x_i\|_p \leq \epsilon} w_i \mathcal{L}(f_\theta(x_i'), y_i),
\tag{5}
$$

where $w_i$ is a weight value assigned for each input sample $(x_i, y_i)$ based on the example size of the class $(x_i, y_i)$ belongs to or some properties of $(x_i, y_i)$. In most existing adversarial training methods (Madry et al., 2017; Zhang et al., 2019; Wang et al., 2019), the cross entropy (CE) loss is adopted as the loss function $\mathcal{L}(\cdot, \cdot)$. However, the CE loss could be suboptimal in imbalanced settings and some new loss functions designed for imbalanced learning specifically, such as Focal loss (Lin et al., 2017) and LDAM loss (Cao et al., 2019), have been proven their superiority in natural training. Hence, besides CE loss, Focal loss and LDAM loss can also be adopted as the loss function $\mathcal{L}(\cdot, \cdot)$ in Eq. (5).

### 4.2 INCREASING FEATURE SEPARABILITY

Our preliminary study indicates that only reweighted adversarial training cannot work well under imbalanced scenarios, and the reweighting strategy in adversarial training behaves very differently from natural training. Hence, in order to facilitate the reweighting strategy in adversarial training under imbalanced scenarios, inspired by our theoretical analysis, we equip a feature separation loss with our SRAT method to enforce the learned feature space as separable as possible. More specifically, the goal of our feature separation loss is to make the learned features of examples from the same class well clustered while from different classes well separated. Hence, adjusting the decision boundary via

the reweighting strategy to fit under-represented classes' examples more will not hurt well-represented classes drastically. The feature separation loss is formally defined as:

$$\mathcal{L}_{sep}(x_i') = -\frac{1}{|P(i)|} \sum_{p \in P(i)} \log \frac{\exp(z_i' \cdot z_p'/\tau)}{\sum_{a \in A(i)} \exp(z_i' \cdot z_a'/\tau)}, \tag{6}$$

where $z_i'$ is the feature representation of the adversarial example $x_i'$ of $x_i$, $\tau \in \mathcal{R}^+$ is a scalar temperature parameter, $P(i)$ denotes the set of input examples belonging to the same class with $x_i$ and $A(i)$ indicates the set of all input examples except $x_i'$. Our proposed feature separation loss $\mathcal{L}_{sep}(\cdot)$ is inspired by the supervised contrastive loss proposed in (Khosla et al., 2020). The main difference is, instead of applying data augmentation techniques to generate two different views of each data example and feeding the model with augmented data examples, our feature separation loss directly takes the adversarial example $x_i'$ of each data example $x_i$ as input.

### 4.3 Training Schedule

By combining the feature separation loss with the reweighted adversarial training, the final object function for Separable Reweighted Adversarial Training (SRAT) is defined as:

$$\min_{\theta} \frac{1}{n} \sum_{i=1}^{n} \max_{\|x_i' - x_i\|_p \leq \epsilon} w_i \mathcal{L}(f_\theta(x_i'), y_i) + \lambda \mathcal{L}_{sep}(x_i'), \tag{7}$$

where $\lambda$ is a hyper-parameter to balance the contributions of two terms.

In practice, in order to better take advantage of the reweighting strategy in our SRAT method, we adopt a deferred reweighting training schedule (Cao et al., 2019). Specifically, before annealing the learning rate, SRAT first trains a model without introducing the reweighting strategy and then applies reweighting into model training process with a smaller learning rate. Since SRAT enables to learn more separable feature space, comparing with applying the reweighting strategy from the beginning of training, this deferred reweighting training schedule enables the reweighting strategy to obtain more benefits from our SRAT method. The training algorithm for SRAT is shown in Appendix A.4.

## 5 Experiment

In this section, we perform experiments to validate the effectiveness of our SRAT method. We first compare SRAT with several representative imbalanced learning methods in adversarial training under various imbalanced scenarios and then conduct ablation study to understand SRAT more deeply.

### 5.1 Experimental Settings

**Datasets.** We conduct experiments on multiple imbalanced training datasets artificially created from three benchmark image datasets CIFAR10, CIFAR100 (Krizhevsky et al., 2009) and SVHN (Netzer et al., 2011) with diverse imbalanced distributions. Specifically, we consider two different imbalance types: Exponential (Exp) imbalance (Cui et al., 2019) and Step imbalance (Buda et al., 2018). For Exp imbalance, the number of training examples of each class will be reduced according to an exponential function $n = n_i \tau^i$, where $i$ is the class index, $n_i$ is the number of training examples in the original training dataset for class $i$ and $\tau \in (0, 1)$. We categorize half classes with most frequent example sizes in the imbalanced training dataset as well-represented classes and the remaining half classes as under-represented classes. For Step imbalance, we follow the similar process adopted in Section 2.1. Moreover, we denote *imbalance ratio* $K$ as the ratio between training example sizes of the most frequent and least frequent class. We construct different imbalanced datasets "Step-10", "Step-100", "Exp-10" and "Exp-100", by adopting different imbalanced types (Step or Exp) with different imbalanced ratios ($K = 10$ or $K = 100$) to train models, and evaluate model's performance on the original uniformly distributed test datasets of CIFAR10, CIFAR100 and SVHN correspondingly. More detailed information about imbalanced training sets can be found in Appendix A.5.

**Baseline methods.** We implement several representative and state-of-the-art imbalanced learning methods (or their combinations) into adversarial training as baseline methods. These methods include: (1) Focal loss (Focal); (2) LDAM loss (LDAM); (3) Class-balanced reweighting (CB-Reweight) (Cui et al., 2019), where each example is reweighted proportionally by the inverse of

Table 1: Performance comparison on the CIFAR10 Step-10 dataset.

| Metric | Standard Accuracy | | Robust Accuracy | |
|---|---|---|---|---|
| Method | Overall | Under | Overall | Under |
| CE | $63.26 \pm 0.59$ | $40.62 \pm 1.10$ | $36.96 \pm 0.36$ | $14.23 \pm 0.83$ |
| Focal | $63.57 \pm 0.92$ | $41.17 \pm 2.07$ | $36.89 \pm 0.36$ | $14.25 \pm 0.97$ |
| LDAM | $57.08 \pm 1.16$ | $31.09 \pm 2.20$ | $37.18 \pm 0.56$ | $12.44 \pm 0.93$ |
| CB-Reweight | $73.30 \pm 0.30$ | $74.80 \pm 0.88$ | $41.34 \pm 0.42$ | $42.15 \pm 1.42$ |
| CB-Focal | $73.42 \pm 0.29$ | $74.35 \pm 1.39$ | $41.34 \pm 0.23$ | $41.80 \pm 1.24$ |
| DRCB-CE | $75.89 \pm 0.23$ | $70.55 \pm 1.10$ | $39.93 \pm 0.24$ | $33.33 \pm 1.42$ |
| DRCB-Focal | $74.61 \pm 0.35$ | $67.06 \pm 1.37$ | $37.91 \pm 0.24$ | $29.50 \pm 1.31$ |
| DRCB-LDAM | $72.95 \pm 0.08$ | $75.42 \pm 1.83$ | $45.23 \pm 0.19$ | $44.98 \pm 1.90$ |
| SRAT-CE | $\mathbf{76.69 \pm 0.33}$ | $73.07 \pm 0.63$ | $41.02 \pm 0.49$ | $36.57 \pm 0.92$ |
| SRAT-Focal | $\underline{75.41 \pm 0.69}$ | $74.91 \pm 0.70$ | $42.05 \pm 0.52$ | $41.28 \pm 0.82$ |
| SRAT-LDAM | $73.99 \pm 0.52$ | $\mathbf{76.63 \pm 0.39}$ | $\mathbf{45.60 \pm 0.18}$ | $\mathbf{45.95 \pm 0.51}$ |

the effective number[2] of its class; (4) Class-balanced Focal loss (CB-Focal) (Cui et al., 2019), a combination of Class-balanced method and Focal loss, where well-classified examples will be downweighted while hard-classified examples will be upweighted controlled by their corresponding effective number; (5) deferred reweighted CE loss (DRCB-CE), where a deferred reweighting training schedule is applied based on the CE loss; (6) deferred reweighted Class-balanced Focal loss (DRCB-Focal), where a deferred reweighting training schedule is applied based on the CB-Focal loss; (7) deferred reweighted Class-balanced LDAM loss (DRCB-LDAM) (Cao et al., 2019), where a deferred reweighting training schedule is applied based on the CB-LDAM loss. In addition, we also include the original PGD adversarial training method using cross entropy loss (CE) in our experiments.

**Our proposed methods.** We evaluate three variants of our proposed SRAT method[3] with different implementations of the prediction loss $\mathcal{L}(\cdot, \cdot)$ in Eq. (5), i.e., CE loss, Focal loss and LDAM loss. The variant utilizing CE loss is denoted as SRAT-CE, and, similarly, other two variants are denoted as SRAT-Focal and SRAT-LDAM, respectively. For all these three variants, Class-balanced method (Cui et al., 2019) is adopted to set weight values within the deferred reweighting training schedule.

**Implementation details.** All aforementioned methods are implemented using a Pytorch library DeepRobust (Li et al., 2020). For CIFAR10/CIFAR100 based datasets, the adversarial examples used in training are calculated by PGD-10, with a perturbation budget $\epsilon = 8/255$ and step size $\gamma = 2/255$; in evaluation, we report robust accuracy under $l_\infty$-norm $8/255$ attacks generated by PGD-20 on Resnet-18 (He et al., 2016) models. For SVHN based datasets, the settings are similar, excepts we set step size $\gamma$ to $1/255$ in both training and evaluation, as suggested in (Wu et al., 2020). We set the total training epochs to 200 and the initial learning rate to 0.1, and decay the learning rate at epoch 160 and 180 with the ratio 0.01. The deferred reweighting strategy will be applied starting from epoch 160.

## 5.2 Performance Comparison

Table 1 and Table 2 show the performance comparison on two different imbalanced CIFAR10 datasets. In these two tables, we use bold values to denote the highest accuracy among all methods and use the underline values to indicate our SRAT variants which achieve the highest accuracy among their corresponding baseline methods utilizing the same loss function for making predictions. Due to the limited space, we report the performance comparison on other imbalanced datasets created from CIFAR10, CIFAR100 and SVHN datasets in Appendix A.6.

From Table 1 and Table 2, we can make the following observations. First, compared to baseline methods, our SRAT method obtains improved performance in terms of both overall standard & robust accuracy under almost all imbalanced settings. More importantly, SRAT makes significant improvements on those under-represented classes, especially under the extremely imbalanced settings. For the CIFAR10 Step-100 dataset, our SRAT-Focal method improves the standard accuracy on under-represented classes from 21.81% achieved by the best baseline method utilizing Focal loss to 51.83% and robust accuracy from 3.24% to 15.89%. These results demonstrate that our SRAT method is able to obtain more robustness under imbalanced settings. Second, the performance gap among

---

[2]The effective number is defined as the volume of examples and can be calculated by $(1 - \beta^{n_i})/(1 - \beta)$, where $\beta \in [0, 1)$ is a hyperparameter and $n_i$ denotes the number of examples of class $i$.

[3]The implementations can be found via `https://github.com/anonymous2share/SRAT`.

Table 2: Performance comparison on the CIFAR10 Step-100 dataset.

| Metric | Standard Accuracy | | Robust Accuracy | |
|---|---|---|---|---|
| Method | Overall | Under | Overall | Under |
| CE | $47.29 \pm 0.32$ | $9.03 \pm 0.99$ | $30.39 \pm 0.24$ | $1.62 \pm 0.41$ |
| Focal | $47.36 \pm 0.19$ | $9.03 \pm 0.52$ | $30.12 \pm 0.31$ | $1.45 \pm 0.12$ |
| LDAM | $42.49 \pm 0.62$ | $0.85 \pm 0.46$ | $30.80 \pm 0.31$ | $0.05 \pm 0.06$ |
| CB-Reweight | $37.68 \pm 1.18$ | $19.64 \pm 1.82$ | $25.58 \pm 0.62$ | $10.33 \pm 0.82$ |
| CB-Focal | $15.44 \pm 3.85$ | $0.00 \pm 0.00$ | $14.46 \pm 3.16$ | $0.00 \pm 0.00$ |
| DRCB-CE | $53.40 \pm 1.20$ | $22.86 \pm 3.03$ | $28.31 \pm 0.59$ | $3.35 \pm 0.56$ |
| DRCB-Focal | $52.75 \pm 0.96$ | $21.81 \pm 2.27$ | $27.78 \pm 0.49$ | $3.24 \pm 0.57$ |
| DRCB-LDAM | $61.60 \pm 0.44$ | $50.69 \pm 2.27$ | $31.37 \pm 0.45$ | $16.25 \pm 2.04$ |
| SRAT-CE | $60.04 \pm 1.16$ | $41.71 \pm 2.07$ | $30.00 \pm 0.80$ | $12.25 \pm 1.43$ |
| SRAT-Focal | $62.93 \pm 1.10$ | $51.83 \pm 3.33$ | $28.38 \pm 1.00$ | $15.89 \pm 3.15$ |
| SRAT-LDAM | $\mathbf{63.13 \pm 1.17}$ | $\mathbf{52.73 \pm 3.23}$ | $\mathbf{33.51 \pm 0.68}$ | $\mathbf{18.89 \pm 0.59}$ |

three SRAT variants are mainly caused by the gap between the loss functions in these methods. As shown in these two tables, DRCB-LDAM typically performs better than DRCE-CE and DRCB-Focal, and similarly, SRAT-LDAM outperforms SRAT-CE and SRAT-Focal under the same settings.

## 5.3 ABLATION STUDY

In this subsection, we provide ablation study to understand our SRAT method more comprehensively.

**Feature space visualization.** In order to facilitate the reweighting strategy in adversarial training under the imbalanced setting, we present a feature separation loss in our SRAT method. The main goal of the feature separation loss is to enforce the learned feature space as much separated as possible. For checking whether the feature separa-

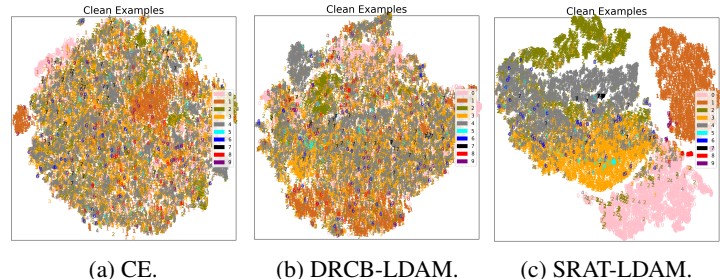

(a) CE.    (b) DRCB-LDAM.    (c) SRAT-LDAM.

Figure 4: t-SNE visualization of feature learned by different methods.

ration loss can work as expected, we apply t-SNE (Van der Maaten & Hinton, 2008) to visualize the latent feature space learned by our SRAT-LDAM method as well as by original PGD adversarial training method (CE) and DRCB-LDAM method in Figure 4.

As shown in Figure 4, the feature space learned by our SRAT-LDAM method is more separable than two baseline methods, which demonstrates that, with our feature separation loss, the adversarially trained model is able to learn much better features and thus SRAT can achieve superiority performance.

**Impact of weight values.** As in all SRAT variants, we adopt the Class-balanced method (Cui et al., 2019) to assign different weights to different classes. To explore how the assigned weights impact the performance of SRAT, we conduct experiments using "Step-100" dataset to see the change of model's performance using different reweighting values. Specifically, we assign well-represented classes with weight 1 and change the weight for under-represented classes from 10 to 200. The experimental results are shown in Figure 5. Here, we use an approximation integer 78 to denote the weight calculated by the Class-balanced method when the imbalance ratio equals 100.

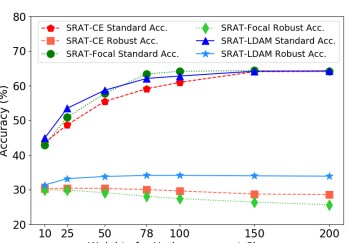

Figure 5: The impact of weights.

From Figure 5, we can obverse that, for all SRAT variants, the model's standard accuracy is increased with the increasing of the weights for under-represented classes. However, the robust accuracy for these three methods do not synchronize with the change of their standard accuracy. When increasing the weights for under-represented classes, robust accuracy of SRAT-LDAM is almost unchanged and of SRAT-CE and SRAT-Focal even has slight decrease. As a trade-off, using a relative large weight, such as 78 or 100, in SRAT can obtain satisfactory performance on both standard & robust accuracy.

**Impact of hyper-parameter** $\lambda$.
In our SRAT method, the contributions of feature separation loss and prediction loss are controlled by a hyper-parameter $\lambda$. In this part, we study how this hyper-parameter affects the performance of SRAT. In experiments, we evaluate the models' performance of all SRAT variants with different values of $\lambda$ used in training process on both "Step-100" and "Exp-100" datasets.

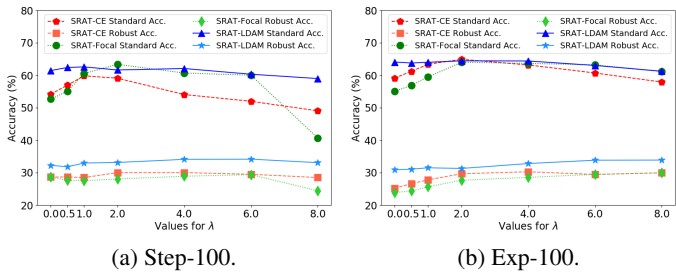

(a) Step-100.      (b) Exp-100.

Figure 6: The impact of the hyper-parameter $\lambda$.

As shown in Figure 6, the performance of all SRAT variants are not very sensitive with the choice of $\lambda$. However, a large value of $\lambda$, such as 8, may hurt the model's performance.

## 6 RELATED WORK

**Adversarial Robustness.** The vulnerability of DNN models to adversarial examples has been verified by many existing successful attack methods (Goodfellow et al., 2014; Carlini & Wagner, 2017). To improve model robustness against adversarial attacks, various defense methods have been proposed (Madry et al., 2017; Raghunathan et al., 2018; Cohen et al., 2019). Among them, adversarial training has been proven to be one of the most effective defense methods (Athalye et al., 2018). Adversarial training can be formulated as solving a min-max optimization problem where the outer minimization process enforces the model to be robust to adversarial examples, generated by the inner maximization process via some existing attacking methods like PGD (Madry et al., 2017). Based on adversarial training, several variants, such as TRADES (Zhang et al., 2019), MART (Wang et al., 2019), have been presented to improve the model's performance further. More details about adversarial robustness can be found in recent surveys (Chakraborty et al., 2018; Xu et al., 2020b). Since almost all studies of adversarial training are focused on balanced datasets, it's worthwhile to investigate the performance of adversarial training methods on imbalanced training datasets.

**Imbalanced Learning.** Most existing works of imbalanced training can be roughly classified into two categories, i.e., re-sampling and reweighting. *Re-sampling* methods aim to reduce imbalance level through either over-sampling examples from under-represented classes (Buda et al., 2018; Byrd & Lipton, 2019) or under-sampling examples from well-represented classes (Japkowicz & Stephen, 2002; Drummond et al., 2003; He & Garcia, 2009). *reweighting* methods allocate different weights for different classes or even different examples. For example, Focal loss (Lin et al., 2017) enlarges the weights of wrongly-classified examples while reducing the weights of well-classified examples in the standard cross entropy loss; and LDAM loss (Cao et al., 2019) regularizes the under-represented classes more strongly than the well-represented classes to attain good generalization on under-represented classes. More details about imbalanced learning can be found in recent surveys (He & Ma, 2013; Johnson & Khoshgoftaar, 2019). The majority of existing methods focused on the nature training scenario and their trained models will be crashed when facing adversarial attacks (Szegedy et al., 2013; Goodfellow et al., 2014). Hence, in this paper, we develop a novel method that can defend adversarial attacks and achieve well-pleasing performance under imbalance settings.

## 7 CONCLUSION

In this work, we first empirically investigate the behavior of adversarial training under imbalanced settings and explore potential solutions to assist adversarial training in tackling the imbalanced issue. As neither adversarial training itself nor adversarial training with reweighting can work well under imbalanced settings, we further theoretically verify that the poor data separability is one key reason causing the failure of adversarial training based methods. Based on our findings, we propose the Separable Reweighted Adversarial Training (SRAT) framework to facilitate the reweighting strategy in imbalanced adversarial training. We validate the effectiveness of SRAT via extensive experiments. In the future, we plan to examine how other types of defense methods perform under imbalanced scenarios and how other types of balanced learning methods behave under adversarial training.

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

# A APPENDIX

In this section, we provide more details about the proposed SRAT framework, as well as the full experimental results of the preliminary study and evaluation.

## A.1 THE BEHAVIOR OF ADVERSARIAL TRAINING

In order to examine the performance of PGD adversarial training under imbalanced scenarios, we adversarially train ResNet18 (He et al., 2016) models on multiple imbalanced training datasets based on CIFAR10 dataset (Krizhevsky et al., 2009). Similar with observations we discussed in Section 2.1, as shown in Figure 7, Figure 8 and Figure 9, adversarial training produces larger performance gap between well-represented classes and under-represented classes than natural training. Especially, in all imbalanced scenarios, adversarially trained models obtain very low robust accuracy on under-represented classes, which proves again that adversarial training cannot be applied in practical imbalanced scenarios directly.

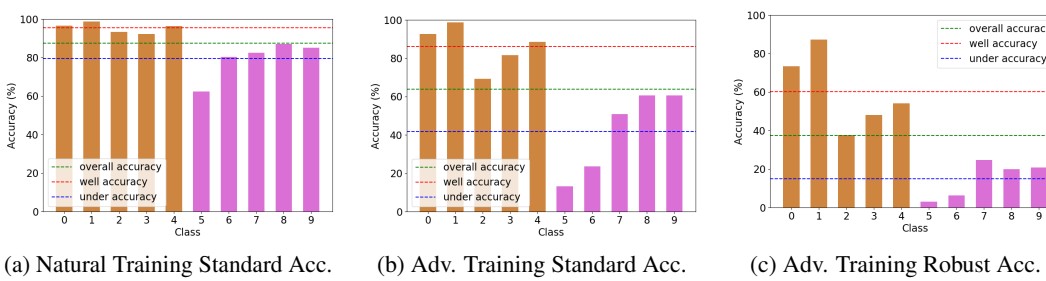

(a) Natural Training Standard Acc.    (b) Adv. Training Standard Acc.    (c) Adv. Training Robust Acc.

Figure 7: Class-wise performance of natural & adversarial training under an imbalanced CIFAR10 dataset "Step-10".

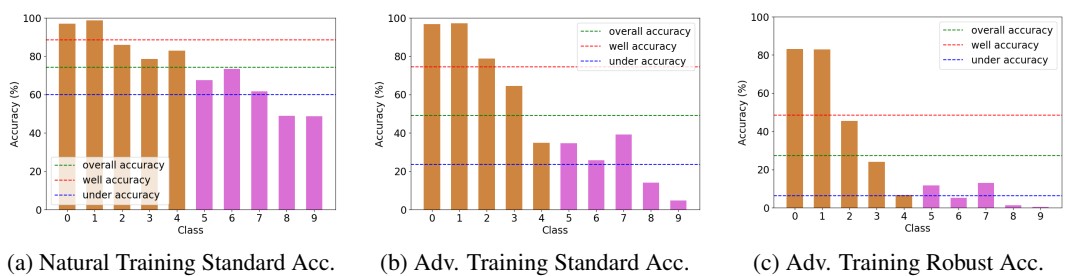

(a) Natural Training Standard Acc.    (b) Adv. Training Standard Acc.    (c) Adv. Training Robust Acc.

Figure 8: Class-wise performance of natural & adversarial training under an imbalanced CIFAR10 dataset "Exp-100".

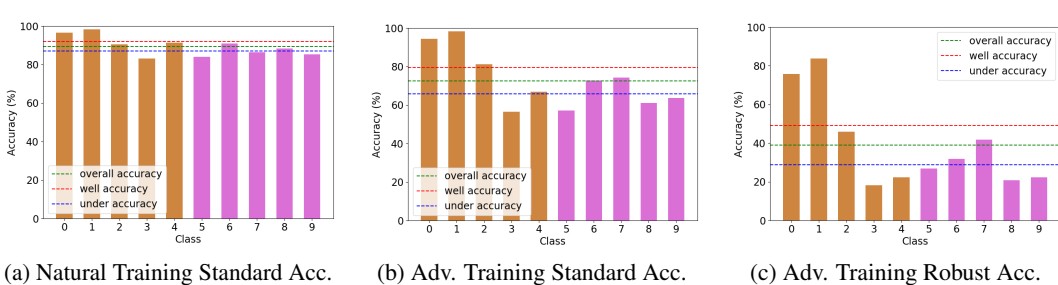

(a) Natural Training Standard Acc.    (b) Adv. Training Standard Acc.    (c) Adv. Training Robust Acc.

Figure 9: Class-wise performance of natural & adversarial training under an imbalanced CIFAR10 dataset "Exp-10".

## A.2 REWEIGHTING STRATEGY IN NATURAL TRAINING V.S. IN ADVERSARIAL TRAINING

For exploring whether the reweighting strategy can help adversarial training deal with imbalanced issues, we evaluate performance of adversarial trained models using diverse binary imbalanced training datasets with different weights assigning to under-represented class. As shown in Figure 10, Figure 11, Figure 12, for adversarially trained models, increasing the weights assigning to under-represented class will improve models' performance on under-represented class. However, as the same time, the models' performance on well-represented class will be drastically decreased. As a comparison, adopting larger weights in naturally trained models will also improve models' performance on under-represented class but only result in slight drop in performance on well-represented class. In other words, the reweighting strategy proposed in natural training to handle imbalanced problem may only provide limited help in adversarial training, and, hence, new techniques are needed for adversarial training under imbalanced scenarios.

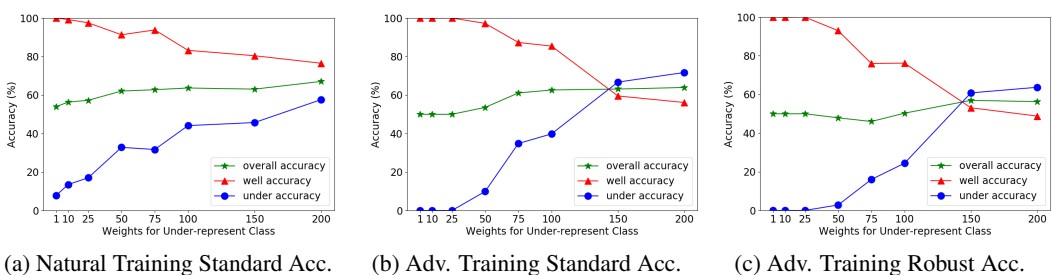

(a) Natural Training Standard Acc.    (b) Adv. Training Standard Acc.    (c) Adv. Training Robust Acc.

Figure 10: Class-wise performance of reweighted natural & adversarial training in binary classification. ("auto" as well-represented class and "truck" as under-represented class).

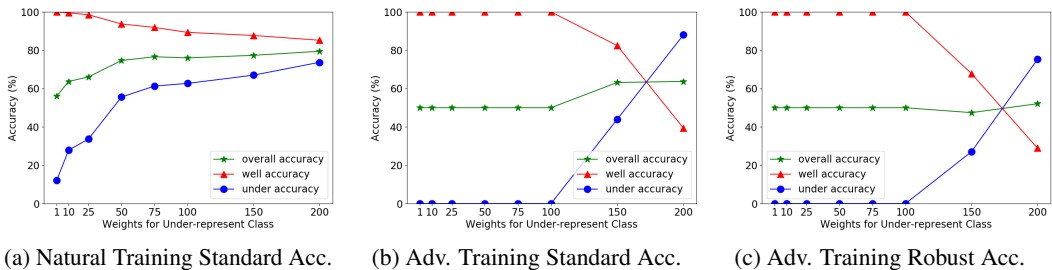

(a) Natural Training Standard Acc.    (b) Adv. Training Standard Acc.    (c) Adv. Training Robust Acc.

Figure 11: Class-wise performance of reweighted natural & adversarial training in binary classification. ("bird" as well-represented class and "frog" as under-represented class).

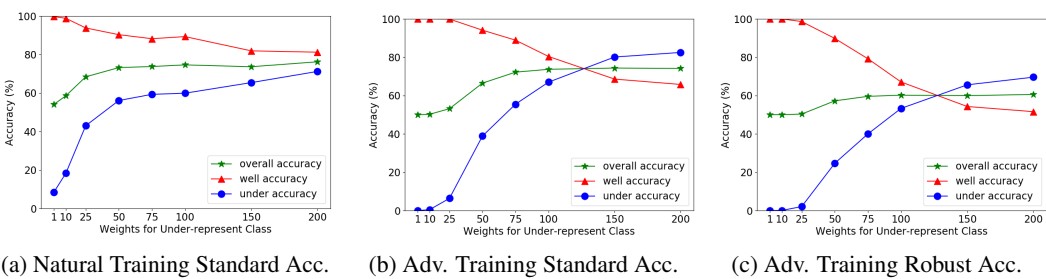

(a) Natural Training Standard Acc.    (b) Adv. Training Standard Acc.    (c) Adv. Training Robust Acc.

Figure 12: Class-wise performance of reweighted natural & adversarial training in binary classification. ("dog" as well-represented class and "deer" as under-represented class).

## A.3 PROOFS OF THE THEOREMS IN SECTION 3

### A.3.1 PROOF OF LEMMA 3.1

**Lemma 3.1** *Under the data distribution $\mathcal{D}$ as defined in Eq. (1), with an imbalanced ratio $K$ and a reweight ratio $\rho$, the optimal classifier which minimizes the (reweighted) empirical risk:*

$$f^* = \arg\min_f \left( Pr.(f(x) \neq y | y = -1) \cdot Pr.(y = -1) \cdot \rho + Pr.(f(x) \neq y | y = +1) \cdot Pr.(y = +1) \right) \quad (2)$$

*has the solution: $w = \mathbf{1}$ and $b = \frac{1}{2}\log(\frac{\rho}{K})\frac{d\sigma^2}{\eta} = \frac{1}{2}\log(\frac{\rho}{K})\frac{d}{S}$.*

**Proof 1 (Proof of Lemma 3.1)** *We will first prove that the optimal model $f^*$ has parameters $w_1 = w_2 = \cdots = w_d$ (or $w = \mathbf{1}$) by contradiction. We define $G = \{1, 2, \ldots, d\}$ and make the following assumption: for the optimal $w$ and $b$, we assume if there exist $w_i < w_j$ for $i \neq j$ and $i, j \in G$. Then we obtain the following standard errors for the class "-1" and the class "+1" of this classifier $f$ with weight $w$:*

$$
\begin{aligned}
Pr.(f^*(x) \neq y | y = -1) &= Pr.(w^T \mathcal{N}(-\eta, \sigma^2) - b > 0) \\
&= Pr.\{ \sum_{k \neq i, k \neq j} w_k \mathcal{N}(-\eta, \sigma^2) + w_i \mathcal{N}(-\eta, \sigma^2) + w_j \mathcal{N}(-\eta, \sigma^2) - b > 0 \}, \\
Pr.(f^*(x) \neq y | y = +1) &= Pr.(w^T \mathcal{N}(+\eta, \sigma^2) - b < 0) \\
&= Pr.\{ \sum_{k \neq i, k \neq j} w_k \mathcal{N}(+\eta, \sigma^2) + w_i \mathcal{N}(+\eta, \sigma^2) + w_j \mathcal{N}(+\eta, \sigma^2) - b < 0 \}.
\end{aligned}
\quad (8)
$$

*However, if we define a new classier $\tilde{f}$ whose weight $\tilde{w}$ uses $w_j$ to replace $w_i$, we obtain the errors for the new classifier:*

$$
\begin{aligned}
Pr.(\tilde{f}(x) \neq y | y = -1) &= Pr.\{ \sum_{k \neq i, k \neq j} w_k \mathcal{N}(-\eta, \sigma^2) + w_j \mathcal{N}(-\eta, \sigma^2) + w_j \mathcal{N}(-\eta, \sigma^2) - b > 0 \}, \\
Pr.(\tilde{f}(x) \neq y | y = +1) &= Pr.\{ \sum_{k \neq i, k \neq j} w_k \mathcal{N}(+\eta, \sigma^2) + w_j \mathcal{N}(+\eta, \sigma^2) + w_j \mathcal{N}(+\eta, \sigma^2) - b < 0 \}.
\end{aligned}
\quad (9)
$$

*Comparing the errors in Eq. (8) and Eq. (9), as $w_i < w_j$, then the classifier $\tilde{f}$ has smaller standard error in each class. Therefore, it contradicts with the assumption that $f$ is the optimal classifier with smallest error. Thus, we conclude for an optimal linear classifier in natural training, it must satisfies $w_1 = w_2 = \cdots = w_d$ (or $w = \mathbf{1}$) if we do not consider the scale of $w$.*

*Next, we calculate the optimal bias term $b$ given $w = \mathbf{1}$, where we find an optimal $b$ can minimize the (reweighted) empirical risk:*

$$
\begin{aligned}
Error_{train}(f^*) &= Pr.(f^*(x) \neq y | y = -1) \cdot Pr.(y = -1) \cdot \rho + Pr.(f^*(x) \neq y | y = +1) \cdot Pr.(y = +1) \\
&\propto Pr.(f^*(x) \neq y | y = -1) \cdot \rho + Pr.(f^*(x) \neq y | y = +1) \cdot K \\
&= \rho \cdot Pr.(\sum_{i=1}^d \mathcal{N}(-\eta, \sigma^2) - b > 0) + K \cdot Pr.(\sum_{i=1}^d \mathcal{N}(\eta, \sigma^2) - b < 0) \\
&= \rho \cdot Pr.(\mathcal{N}(0,1) < -\frac{b + d\eta}{d\sigma}) + K \cdot Pr.(\mathcal{N}(0,1) < \frac{b - d\eta}{d\sigma}),
\end{aligned}
$$

*and we take the derivative with respect to $b$:*

$$\frac{\partial Error_{train}}{\partial b} = \frac{\rho}{\sqrt{2\pi}} \cdot (-\frac{1}{d\sigma}) \exp(-\frac{1}{2}(-\frac{b + d\eta}{d\sigma})^2) + \frac{K}{\sqrt{2\pi}} \cdot (\frac{1}{d\sigma}) \exp(-\frac{1}{2}(\frac{b - d\eta}{d\sigma})^2).$$

*When $\partial Error_{train}/\partial b = 0$, we can calculate the optimal $b$ which gives the minimum value of the empirical error, and we have:*

$$b = \frac{1}{2}\log(\frac{\rho}{K})\frac{d\sigma^2}{\eta} = \frac{1}{2}\log(\frac{\rho}{K})\frac{d}{S}.$$

### A.3.2 Proof of Theorem 3.1

**Theorem 3.1** *Under two data distributions $(x^{(1)}, y^{(1)}) \in \mathcal{D}_1$ and $(x^{(2)}, y^{(2)}) \in \mathcal{D}_2$ with different separabilities $S_1 > S_2$, let $f_1^*$ and $f_2^*$ be the optimal non-reweighted classifiers ($\rho = 1$) under $\mathcal{D}_1$ and $\mathcal{D}_2$, respectively. Given the imbalance ratio $K$ is large enough, we have:*

$$
\begin{aligned}
&Pr.(f_1^*(x^{(1)}) \neq y^{(1)}|y^{(1)} = -1) - Pr.(f_1^*(x^{(1)}) \neq y^{(1)}|y^{(1)} = +1) \\
&< Pr.(f_2^*(x^{(2)}) \neq y^{(2)}|y^{(2)} = -1) - Pr.(f_2^*(x^{(2)}) \neq y^{(2)}|y^{(2)} = +1).
\end{aligned}
\tag{3}
$$

**Proof 2 (Proof of Theorem 3.1)** *Without loss of generality, for distribution $\mathcal{D}_1, \mathcal{D}_2$ with different mean-variance pairs $(\pm\eta_1, \sigma_1^2)$ and $(\pm\eta_2, \sigma_2^2)$, we can only consider the case $\eta_1 = \eta_2$ and $\sigma_1^2 < \sigma_2^2$. Otherwise, we can simply rescale one of them to match the mean vector of the other and will not impact the results. Under this definition, the optimal classifier $f_1^*$ and $f_2^*$ has weight vector $w_1 = w_2 = \mathbf{1}$ and bias term $b_1, b_2$, with the value as demonstrated in Lemma 3.1. Next, we will prove the Theorem 3.1 by 2 steps.*

*Step 1. For the error of class "-1", we have:*

$$
\begin{aligned}
Pr.(f_1^*(x^{(1)}) \neq y^{(1)}|y^{(1)} = -1) &= Pr.(\sum_{i=1}^{d} \mathcal{N}(-\eta, \sigma_1^2) - b_1 > 0) \\
&< Pr.(\sum_{i=1}^{d} \mathcal{N}(-\eta, \sigma_1^2) - b_2 > 0) \quad (\text{because } S_1 > S_2) \\
&< Pr.(\sum_{i=1}^{d} \mathcal{N}(-\eta, \sigma_2^2) - b_2 > 0) \quad (\text{because } \sigma_1^2 < \sigma_2^2) \\
&= Pr.(f_2^*(x^{(2)}) \neq y^{(2)}|y^{(2)} = -1).
\end{aligned}
$$

*Step 2. For the error of class "+1", we have:*

$$
\begin{aligned}
Pr.(f_1^*(x^{(1)}) \neq y^{(1)}|y^{(1)} = +1) &= Pr.(\sum_{i=1}^{d} \mathcal{N}(\eta, \sigma_1^2) - b_1 < 0) \\
&= Pr.(\mathcal{N}(0, 1) < \frac{b_1 - d\eta}{d\sigma_1}) \\
&= Pr.(\mathcal{N}(0, 1) < \frac{-\log(K) \cdot \sigma_1}{2\eta} - \frac{\eta}{\sigma_1}),
\end{aligned}
\tag{10}
$$

*and similarly,*

$$
Pr.(f_2^*(x^{(2)}) \neq y^{(2)}|y^{(2)} = +1) = Pr.(\mathcal{N}(0, 1) < \frac{-\log(K) \cdot \sigma_2}{2\eta} - \frac{\eta}{\sigma_2}).
\tag{11}
$$

*Note that when $K$ is large enough, i.e., $\log(K) > \frac{2 \cdot \eta^2}{\sigma_1 \cdot \sigma_2}$, we can get the Z-score in Eq. (10) is larger than Eq. (11). As a result, we have:*

$$
Pr.(f_1^*(x^{(1)}) \neq y^{(1)}|y^{(1)} = +1) > Pr.(f_2^*(x^{(2)}) \neq y^{(2)}|y^{(2)} = +1).
\tag{12}
$$

*By combining Step 1 and Step 2, we can get the inequality in Theorem 3.1.*

### A.3.3 Proof of Theorem 3.2

**Theorem 3.2** *Under two data distributions $(x^{(1)}, y^{(1)}) \in \mathcal{D}_1$ and $(x^{(2)}, y^{(2)}) \in \mathcal{D}_2$ with different separabilities $S_1 > S_2$, let $f_1^*$ and $f_2^*$ be the optimal non-reweighted classifiers ($\rho = 1$) under $\mathcal{D}_1$ and $\mathcal{D}_2$, respectively, and let $f_1'^*$ and $f_2'^*$ be the optimal reweighted classifiers under $\mathcal{D}_1$ and $\mathcal{D}_2$ given the optimal reweighting ratio ($\rho = K$). Given the imbalance ratio $K$ is large enough, we have:*

$$
\begin{aligned}
&Pr.(f_1'^*(x^{(1)}) \neq y^{(1)}|y^{(1)} = +1) - Pr.(f_1^*(x^{(1)}) \neq y^{(1)}|y^{(1)} = +1) \\
&< Pr.(f_2'^*(x^{(2)}) \neq y^{(2)}|y^{(2)} = +1) - Pr.(f_2^*(x^{(2)}) \neq y^{(2)}|y^{(2)} = +1).
\end{aligned}
\tag{4}
$$

**Proof 3 (Proof of Theorem 3.2)** *We first show that under both distribution $\mathcal{D}_1$ and $\mathcal{D}_2$, the optimal reweighting ratio $\rho$ is equal to the imbalance ratio $K$. Based on the results in Eq. (8) and calculated model parameters $w$ and $b$, we have the test error (given the model trained by reweight value $\rho$):*

$$
\begin{aligned}
&Error_{test}(f^*) \\
&= Pr.(f^*(x) \neq y | y = -1) \cdot Pr.(y = -1) + Pr.(f^*(x) \neq y | y = +1) \cdot Pr.(y = +1) \\
&\propto Pr.(\mathcal{N}(0,1) < -\frac{b + d\eta}{d\sigma}) + Pr.(\mathcal{N}(0,1) < \frac{b - d\eta}{d\sigma}) \\
&= Pr.(\mathcal{N}(0,1) < -\frac{1}{2}\log(\frac{\rho}{K}) - \frac{\sigma}{\eta}) + Pr.(\mathcal{N}(0,1) < \frac{1}{2}\log(\frac{\rho}{K}) - \frac{\sigma}{\eta}).
\end{aligned}
$$

*The value of taking the minimum when its derivative with respect to $\rho$ is equal to $0$, where we can get $\rho = K$ and the bias term $b = 0$. Note that the variance values have the relation: $\sigma_1^2 < \sigma_2^2$. Therefore, it is easy to get that:*

$$
\begin{aligned}
Pr.(f_1'^*(x^{(1)}) \neq y^{(1)} | y^{(1)} = +1) &= Pr.(\sum_{i=1}^{d} \mathcal{N}(\eta, \sigma_1^2) < 0) \\
&< Pr.(\sum_{i=1}^{d} \mathcal{N}(\eta, \sigma_2^2) < 0) = Pr.(f_2'^*(x^{(2)}) \neq y^{(2)} | y^{(2)} = +1).
\end{aligned}
\tag{13}
$$

*Combining the results in Eq. (12) and (13), we have proved the inequality in Theorem 3.2.*

### A.4 ALGORITHM OF SRAT

The algorithm of our proposed SRAT framework is shown in Algorithm 1. Specifically, in each training iteration, we first generate adversarial examples using PGD for examples in the current batch (Line 5). If the current training iteration does not reach a predefined starting reweighting epoch $T_d$, we will assign same weights, i.e., $w_i = 1$ for all adversarial examples $x_i$ in the current batch (Line 6). Otherwise, the reweighting strategy will be adopted in the final loss function (Line 15), where a specific weight $w_i$ will be assigned for each adversarial example $x_i$ if its corresponding clean example $x_i$ comes from an under-represented class.

---

**Algorithm 1** Separable Reweighted Adversarial Training (SRAT).

---

**Input:** imbalanced training dataset $D = \{(x_i, y_i)\}_{i=1}^{n}$, number of total training epochs $T$, starting reweighting epoch $T_d$, batch size $N$, number of batches $M$, learning rate $\gamma$
**Output:** An adversarially robust model $f_\theta$
1: Initialize the model parameters $\theta$ randomly;
2: **for** epoch $= 1, \ldots, T_d - 1$ **do**
3:     **for** mini-batch $= 1, \ldots, M$ **do**
4:         Sample a mini-batch $\mathcal{B} = \{(x_i, y_i)\}_{i=1}^{N}$ from $D$;
5:         Generate adversarial example $x_i'$ for each $x_i' \in \mathcal{B}$;
6:         $\mathcal{L}(f_\theta) = \frac{1}{N}\sum_{i=1}^{N}\max_{\|x_i'-x_i\|_p \leq \epsilon}\mathcal{L}(f_\theta(x_i'), y_i) + \lambda\mathcal{L}_{sep}(x_i')$
7:         $\theta \leftarrow \theta - \gamma\nabla_\theta\mathcal{L}(f_\theta)$
8:     **end for**
9:     Optional: $\gamma \leftarrow \gamma/\kappa$
10: **end for**
11: **for** epoch $= T_d, \ldots, T$ **do**
12:     **for** mini-batch $= 1, \ldots, M$ **do**
13:         Sample a mini-batch $\mathcal{B} = \{(x_i, y_i)\}_{i=1}^{N}$ from $D$;
14:         Generate adversarial example $x_i'$ for each $x_i' \in \mathcal{B}$;
15:         $\mathcal{L}(f_\theta) = \frac{1}{N}\sum_{i=1}^{N}\max_{\|x_i'-x_i\|_p \leq \epsilon} w_i\mathcal{L}(f_\theta(x_i'), y_i) + \lambda\mathcal{L}_{sep}(x_i')$
16:         $\theta \leftarrow \theta - \gamma\nabla_\theta\mathcal{L}(f_\theta)$
17:     **end for**
18:     Optional: $\gamma \leftarrow \gamma/\kappa$
19: **end for**

---

## A.5  Data Distribution of Imbalanced Training Datasets

In our experiments, we construct multiple imbalanced training datasets to simulate various kinds of imbalanced scenarios by combining different imbalance types (i.e., Exp and Step) with different imbalanced ratios (i.e., $K = 10$ and $K = 100$). Figure 13 and Figure 14 show the data distribution of all ten-classes imbalanced training datasets used in our preliminary studies and experiments based on CIFAR10 (Krizhevsky et al., 2009) and SVHN (Netzer et al., 2011) datasets, respectively.

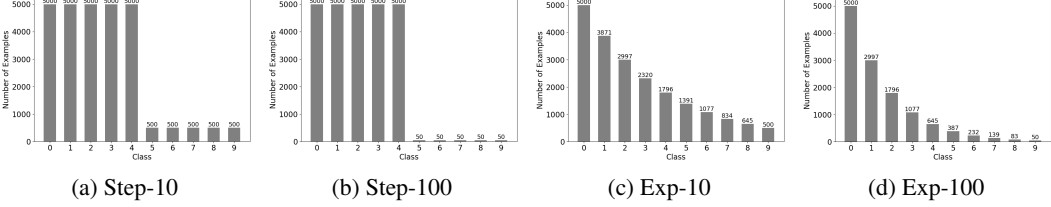

|  (a) Step-10  |  (b) Step-100  |  (c) Exp-10  |  (d) Exp-100  |

Figure 13: Data distribution of imbalanced training datasets constructed from CIFAR10 dataset.

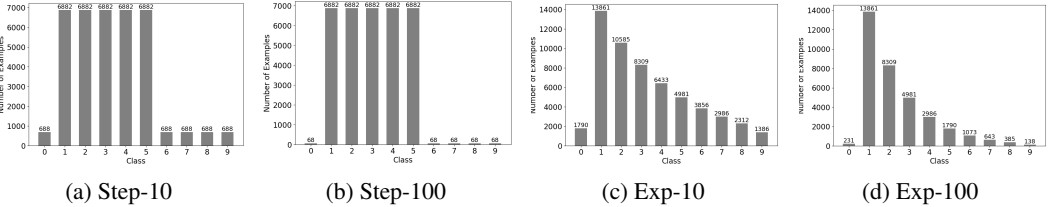

|  (a) Step-10  |  (b) Step-100  |  (c) Exp-10  |  (d) Exp-100  |

Figure 14: Data distribution of imbalanced training datasets constructed from SVHN dataset.

## A.6  Performance Comparison

### A.6.1  Performance Comparison on Imbalanced CIFAR10 Datasets

Table 3 and Table 4 show the performance comparison on two imbalanced CIFAR10 datasets. We use bold values to denote the highest accuracy among all methods and use the underline values to indicate our SRAT variants which achieve the highest accuracy among their corresponding baseline methods utilizing the same loss function for making predictions.

From Table 3 and Table 4, we get similar observation that, comparing with baseline methods, our proposed SRAT method can produce a robust model which can achieve improved overall performance when the training dataset is imbalanced. In addition, based on the experimental results in Table 3 to Table 4, we find that, compared with the performance improvement between DRCB-LDAM and SRAT-LDAM, the improvement between DRCB-CE and SRAT-CE and the improvement between DRCB-Focal and SRAT-Focal are more obviously. The possible reason behind this phenomenon is, the LDAM loss can also implicitly produce a more separable feature space (Cao et al., 2019) while CE loss and Focal loss do not conduct any specific operations on the latent feature space. Hence, the feature separation loss contained in SRAT-CE and SRAT-Focal could be more effective on learning separable feature space and facilitate the Focal loss on prediction. However, in SRAT-LDAM, the feature separation loss and LDAM loss may affect each other on learning feature representations and, hence, the effectiveness of the feature separation loss may be counteracted or weakened.

### A.6.2  Performance Comparison on Imbalanced SVHN Datasets

We report the performance comparison on various imbalanced SVHN datasets with different imbalance types and imbalance ratios from Table 5 to Table 8.

Based on experimental results shown in Table 5 and Table 8, we find that the robust model trained by our SRAT method can achieve superior overall performance under various imbalanced scenarios, when comparing with their corresponding baselines methods which utilize the same prediction loss function. Hence, experiments conducted on various imbalanced SVHN datasets further verify the effectiveness of our SRAT method under diverse imbalanced scenarios.

Table 3: Performance comparison on the CIFAR10 Exp-10 dataset.

| Metric | Standard Accuracy | | Robust Accuracy | |
|---|---|---|---|---|
| Method | Overall | Under | Overall | Under |
| CE | $71.95 \pm 0.52$ | $64.09 \pm 0.44$ | $37.94 \pm 0.19$ | $26.79 \pm 0.51$ |
| Focal | $72.06 \pm 0.78$ | $63.99 \pm 1.15$ | $37.62 \pm 0.34$ | $26.27 \pm 1.04$ |
| LDAM | $67.39 \pm 1.00$ | $58.01 \pm 2.26$ | $41.35 \pm 0.32$ | $28.65 \pm 0.83$ |
| CB-Reweight | $75.17 \pm 0.15$ | $76.87 \pm 0.69$ | $41.02 \pm 0.39$ | $41.67 \pm 0.89$ |
| CB-Focal | $74.73 \pm 0.41$ | $76.67 \pm 0.26$ | $38.86 \pm 0.67$ | $42.41 \pm 0.56$ |
| DRCB-CE | $76.25 \pm 0.09$ | $75.83 \pm 0.49$ | $40.02 \pm 0.45$ | $37.93 \pm 0.65$ |
| DRCB-Focal | $75.36 \pm 0.40$ | $72.72 \pm 0.94$ | $37.76 \pm 0.54$ | $33.83 \pm 0.68$ |
| DRCB-LDAM | $73.92 \pm 0.31$ | $78.53 \pm 1.24$ | $46.29 \pm 0.46$ | $48.81 \pm 0.54$ |
| SRAT-CE | $\mathbf{76.74 \pm 0.15}$ | $78.61 \pm 0.63$ | $42.39 \pm 0.71$ | $43.37 \pm 0.38$ |
| SRAT-Focal | $75.26 \pm 0.00$ | $\mathbf{80.52 \pm 0.00}$ | $42.37 \pm 0.00$ | $47.22 \pm 0.00$ |
| SRAT-LDAM | $74.63 \pm 0.00$ | $79.82 \pm 0.00$ | $\mathbf{46.72 \pm 0.00}$ | $\mathbf{50.38 \pm 0.00}$ |

Table 4: Performance comparison on the CIFAR10 Exp-100 dataset.

| Metric | Standard Accuracy | | Robust Accuracy | |
|---|---|---|---|---|
| Method | Overall | Under | Overall | Under |
| CE | $48.40 \pm 0.59$ | $23.04 \pm 1.15$ | $26.94 \pm 0.84$ | $6.17 \pm 0.86$ |
| Focal | $49.16 \pm 0.61$ | $23.69 \pm 1.15$ | $26.84 \pm 0.59$ | $5.88 \pm 0.48$ |
| LDAM | $48.39 \pm 0.99$ | $25.69 \pm 1.35$ | $29.51 \pm 0.27$ | $8.95 \pm 0.45$ |
| CB-Reweight | $57.49 \pm 0.58$ | $56.47 \pm 1.67$ | $29.01 \pm 0.30$ | $26.53 \pm 1.27$ |
| CB-Focal | $50.35 \pm 0.44$ | $60.05 \pm 0.53$ | $27.15 \pm 0.20$ | $\mathbf{33.56 \pm 0.35}$ |
| DRCB-CE | $57.30 \pm 0.30$ | $37.90 \pm 1.23$ | $26.97 \pm 0.55$ | $10.57 \pm 1.03$ |
| DRCB-Focal | $54.76 \pm 0.30$ | $31.79 \pm 1.30$ | $25.24 \pm 0.39$ | $7.81 \pm 0.87$ |
| DRCB-LDAM | $62.65 \pm 0.50$ | $57.19 \pm 2.10$ | $31.66 \pm 0.56$ | $22.11 \pm 1.70$ |
| SRAT-CE | $\mathbf{64.29 \pm 0.46}$ | $61.81 \pm 1.83$ | $29.99 \pm 0.43$ | $24.09 \pm 0.98$ |
| SRAT-Focal | $62.57 \pm 0.47$ | $64.88 \pm 0.81$ | $30.34 \pm 0.67$ | $28.66 \pm 1.60$ |
| SRAT-LDAM | $63.11 \pm 0.08$ | $\mathbf{65.60 \pm 1.94}$ | $\mathbf{34.22 \pm 0.41}$ | $32.55 \pm 1.70$ |

### A.6.3 PERFORMANCE COMPARISON ON IMBALANCED CIFAR100 DTASETS

Table 9 and Table 10 show the performance comparison between our SRAT variants with baseline methods on two imbalanced CIFAR100 datasets.

As shown in Table 9 and Table 10, although the increasing of total number of classes in the dataset brings more challenges on the model training process, our SRAT variants are able to achieve notable improvements on both standard accuracy and robust accuracy.

Therefore, in conclusion, experiments conducted on multiple imbalanced datasets created from three benchmark image datasets CIFAR10, CIFAR100 (Krizhevsky et al., 2009) and SVHN (Netzer et al., 2011) with diverse imbalanced distributions verify the effectiveness of our proposed SRAT method under various imbalanced scenarios.

## A.7 ABLATION STUDY

**Impact of weight values.** We study the impact of weight values in Section 5.3 via evaluating the overall standard accuracy and robust accuracy of SRAT trained models with different weights assigning to under-represented classes. Here, we further investigate the impact of weight values in our SART method from the class-wise perspective. In Figure 15, we report the standard & robust accuracy of models trained by our SRAT-LDAM variant with

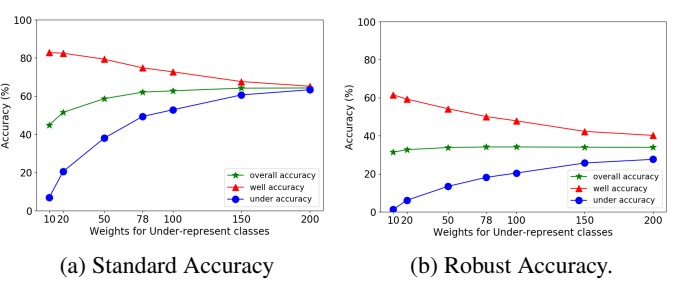

(a) Standard Accuracy      (b) Robust Accuracy.

Figure 15: The impact of weights for the SRAT-LDAM method.

Table 5: Performance comparison on the SVHN Step-10 dataset.

| Metric | Standard Accuracy | | Robust Accuracy | |
|---|---|---|---|---|
| Method | Overall | Under | Overall | Under |
| CE | $79.88 \pm 0.17$ | $67.04 \pm 0.37$ | $37.62 \pm 0.10$ | $22.08 \pm 0.26$ |
| Focal | $79.96 \pm 0.15$ | $67.03 \pm 0.38$ | $37.83 \pm 0.17$ | $22.47 \pm 0.43$ |
| LDAM | $84.55 \pm 0.83$ | $74.96 \pm 1.52$ | $45.80 \pm 0.31$ | $31.23 \pm 0.70$ |
| CB-Reweight | $79.48 \pm 0.27$ | $66.07 \pm 0.42$ | $37.38 \pm 0.20$ | $21.66 \pm 0.26$ |
| CB-Focal | $80.29 \pm 0.15$ | $67.56 \pm 0.23$ | $38.10 \pm 0.24$ | $23.00 \pm 0.38$ |
| DRCB-CE | $80.62 \pm 0.25$ | $68.74 \pm 0.50$ | $37.25 \pm 0.15$ | $22.79 \pm 0.42$ |
| DRCB-Focal | $79.11 \pm 0.15$ | $65.72 \pm 0.44$ | $37.01 \pm 0.33$ | $22.02 \pm 0.55$ |
| DRCB-LDAM | $\mathbf{87.83 \pm 0.68}$ | $\mathbf{82.63 \pm 1.17}$ | $\mathbf{46.45 \pm 0.30}$ | $\mathbf{35.15 \pm 0.69}$ |
| SRAT-CE | $82.89 \pm 0.91$ | $72.79 \pm 1.93$ | $38.23 \pm 0.78$ | $24.70 \pm 1.62$ |
| SRAT-Focal | $85.32 \pm 0.19$ | $77.75 \pm 0.51$ | $39.53 \pm 0.40$ | $28.41 \pm 0.99$ |
| SRAT-LDAM | $87.65 \pm 0.09$ | $82.62 \pm 0.25$ | $46.03 \pm 0.10$ | $34.75 \pm 0.48$ |

Table 6: Performance comparison on the SVHN Step-100 dataset.

| Metric | Standard Accuracy | | Robust Accuracy | |
|---|---|---|---|---|
| Method | Overall | Under | Overall | Under |
| CE | $59.61 \pm 1.59$ | $26.19 \pm 3.08$ | $29.57 \pm 0.59$ | $5.03 \pm 1.49$ |
| Focal | $60.58 \pm 1.40$ | $28.17 \pm 2.76$ | $30.27 \pm 0.51$ | $5.83 \pm 1.42$ |
| LDAM | $65.61 \pm 0.43$ | $37.13 \pm 0.71$ | $33.34 \pm 0.07$ | $8.36 \pm 0.54$ |
| CB-Reweight | $60.23 \pm 1.74$ | $27.68 \pm 3.49$ | $29.54 \pm 0.76$ | $5.32 \pm 1.40$ |
| CB-Focal | $60.73 \pm 0.40$ | $28.37 \pm 0.63$ | $30.09 \pm 0.09$ | $5.75 \pm 0.61$ |
| DRCB-CE | $60.67 \pm 1.04$ | $28.36 \pm 2.08$ | $30.02 \pm 1.04$ | $5.59 \pm 1.30$ |
| DRCB-Focal | $61.65 \pm 1.21$ | $30.29 \pm 2.02$ | $30.78 \pm 0.32$ | $7.06 \pm 0.93$ |
| DRCB-LDAM | $63.78 \pm 1.79$ | $33.99 \pm 3.50$ | $33.60 \pm 0.23$ | $7.28 \pm 1.47$ |
| SRAT-CE | $63.39 \pm 0.64$ | $33.85 \pm 1.11$ | $29.64 \pm 0.21$ | $6.11 \pm 0.29$ |
| SRAT-Focal | $69.27 \pm 1.38$ | $45.50 \pm 3.16$ | $31.58 \pm 0.70$ | $9.87 \pm 1.61$ |
| SRAT-LDAM | $\mathbf{71.56 \pm 1.25}$ | $\mathbf{50.33 \pm 2.29}$ | $\mathbf{33.54 \pm 0.52}$ | $\mathbf{11.63 \pm 1.09}$ |

different weight values on all ten classes, five well-represented classes and five under-represented classes of the Step-100 CIFAR10 dataset separately. Same as before, we assign well-represented classes with weight 1 and change the weight for under-represented classes from 10 to 200. In addition, we use an approximation integer 78 to denote the weight calculated by the Class-balanced method (Cui et al., 2019).

As shown in Figure 15, when increasing of the weights for under-represented classes, the model's performance on under-represented classes can be increased greatly, with the cost of relatively small performance dropping on well-represented classes, and the overall performance of the model is also increased. This phenomenon can be observed much clear in the standard accuracy case, as shown in Figure 15a. Recall that in our preliminary study (Section 2), we find the reweighting strategy cannot work well in adversarial training, as it causes a strong tension of the model's performance between well-represented and under-represented class. As this strong tension does not exist in the model trained by our SRAT method, we believe our SRAT method indeed facilitate the reweighting strategy in adversarial training under imbalanced scenarios.

**Impact of imbalance ratio $K$.** In previous experiments, we evaluate the effectiveness of our SRAT method using various imbalanced datasets with imbalance ratio $K = 10$ or $K = 100$. To investigate the performance of our SRAT method more comprehensively, in this part, we test our SRAT method on more imbalanced datasets with diverse imbalance ratios. Specifically, we construct a series of "Step" imbalanced CIFAR10 datasets by setting the value of the imbalance ratio $K$ from 5 to 100. For comparison, we apply both DRCB-Focal method and our SRAT-Focal variant to train models on those imbalanced datasets and test the trained models' performance on the original uniformly distributed CIFAR10 test dataset. The experimental results are shown in Figure 16.

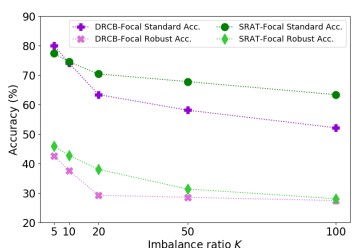

Figure 16: The impact of imbalance ratio $K$.

Table 7: Performance comparison on the SVHN Exp-10 dataset.

| Metric | Standard Accuracy | | Robust Accuracy | |
|---|---|---|---|---|
| Method | Overall | Under | Overall | Under |
| CE | $87.54 \pm 0.61$ | $82.67 \pm 1.01$ | $44.12 \pm 0.36$ | $35.33 \pm 0.19$ |
| Focal | $87.82 \pm 0.52$ | $83.01 \pm 0.90$ | $44.88 \pm 0.55$ | $35.97 \pm 0.09$ |
| LDAM | $90.06 \pm 0.59$ | $86.69 \pm 0.70$ | $51.84 \pm 0.68$ | $43.73 \pm 0.97$ |
| CB-Reweight | $87.66 \pm 0.59$ | $82.79 \pm 0.99$ | $44.39 \pm 0.57$ | $35.53 \pm 0.53$ |
| CB-Focal | $87.86 \pm 0.53$ | $82.96 \pm 0.99$ | $44.61 \pm 0.48$ | $35.55 \pm 0.35$ |
| DRCB-CE | $88.49 \pm 0.55$ | $84.51 \pm 0.89$ | $43.82 \pm 0.46$ | $36.28 \pm 0.37$ |
| DRCB-Focal | $87.47 \pm 0.48$ | $82.78 \pm 0.72$ | $42.52 \pm 0.60$ | $34.31 \pm 0.54$ |
| DRCB-LDAM | $91.24 \pm 0.57$ | $\mathbf{89.65 \pm 0.70}$ | $\mathbf{52.39 \pm 0.74}$ | $\mathbf{46.71 \pm 1.46}$ |
| SRAT-CE | $88.70 \pm 0.37$ | $84.94 \pm 0.40$ | $44.54 \pm 0.66$ | $36.59 \pm 0.68$ |
| SRAT-Focal | $89.36 \pm 0.44$ | $85.93 \pm 0.44$ | $45.41 \pm 0.55$ | $38.18 \pm 0.79$ |
| SRAT-LDAM | $\mathbf{91.27 \pm 0.46}$ | $89.55 \pm 0.66$ | $52.10 \pm 0.85$ | $46.13 \pm 1.23$ |

Table 8: Performance comparison on the SVHN Exp-100 dataset.

| Metric | Standard Accuracy | | Robust Accuracy | |
|---|---|---|---|---|
| Method | Overall | Under | Overall | Under |
| CE | $72.51 \pm 0.46$ | $56.30 \pm 0.85$ | $33.34 \pm 0.42$ | $16.93 \pm 0.24$ |
| Focal | $72.61 \pm 0.37$ | $56.48 \pm 0.57$ | $34.09 \pm 0.37$ | $17.62 \pm 0.29$ |
| LDAM | $79.11 \pm 0.74$ | $66.86 \pm 1.14$ | $\mathbf{40.42 \pm 0.75}$ | $\mathbf{25.18 \pm 1.29}$ |
| CB-Reweight | $72.25 \pm 0.45$ | $55.97 \pm 0.84$ | $33.36 \pm 0.40$ | $17.16 \pm 0.77$ |
| CB-Focal | $73.23 \pm 0.50$ | $57.34 \pm 0.96$ | $34.25 \pm 0.37$ | $17.90 \pm 0.53$ |
| DRCB-CE | $73.74 \pm 0.53$ | $58.03 \pm 1.14$ | $33.52 \pm 0.13$ | $17.68 \pm 0.40$ |
| DRCB-Focal | $71.95 \pm 0.09$ | $55.11 \pm 0.22$ | $33.43 \pm 0.36$ | $17.63 \pm 0.49$ |
| DRCB-LDAM | $80.29 \pm 0.28$ | $69.23 \pm 0.12$ | $40.16 \pm 0.74$ | $24.64 \pm 0.77$ |
| SRAT-CE | $77.11 \pm 0.48$ | $64.47 \pm 1.19$ | $34.48 \pm 0.19$ | $19.91 \pm 0.68$ |
| SRAT-Focal | $\mathbf{81.30 \pm 0.91}$ | $\mathbf{72.26 \pm 2.00}$ | $36.71 \pm 0.53$ | $24.84 \pm 1.59$ |
| SRAT-LDAM | $80.71 \pm 0.40$ | $70.49 \pm 0.71$ | $40.33 \pm 0.43$ | $25.11 \pm 0.58$ |

From Figure 16, we can obverse that, under different imbalanced scenarios, the model trained by our SRAT-Focal can always achieve better performance than the one trained by DRCB-Focal method. In other words, the effectiveness of our SRAT method will not be affected by the imbalanced ratio $K$, which determines the data distribution of the imbalanced training dataset.

Table 9: Performance comparison on the CIFAR100 Step-10 dataset.

| Metric | Standard Accuracy | | Robust Accuracy | |
|---|---|---|---|---|
| Method | Overall | Under | Overall | Under |
| CE | $39.90 \pm 0.11$ | $17.90 \pm 0.38$ | $17.88 \pm 0.32$ | $6.40 \pm 0.60$ |
| Focal | $40.10 \pm 0.27$ | $17.99 \pm 0.75$ | $17.67 \pm 0.30$ | $6.40 \pm 0.18$ |
| LDAM | $39.34 \pm 0.54$ | $17.57 \pm 0.94$ | $20.95 \pm 0.20$ | $7.41 \pm 0.37$ |
| DRCB-CE | $45.21 \pm 0.11$ | $33.26 \pm 0.09$ | $18.36 \pm 0.33$ | $11.15 \pm 0.48$ |
| DRCB-Focal | $44.28 \pm 0.15$ | $30.57 \pm 0.22$ | $17.30 \pm 0.39$ | $9.73 \pm 0.18$ |
| DRCB-LDAM | $44.70 \pm 0.46$ | $35.90 \pm 0.92$ | $21.80 \pm 0.12$ | $15.19 \pm 0.36$ |
| SRAT-CE | $\mathbf{47.17 \pm 0.26}$ | $37.81 \pm 0.38$ | $21.36 \pm 0.31$ | $15.41 \pm 0.19$ |
| SRAT-Focal | $\underline{46.83 \pm 0.28}$ | $\mathbf{38.10 \pm 0.58}$ | $\underline{21.66 \pm 0.32}$ | $16.52 \pm 0.32$ |
| SRAT-LDAM | $45.41 \pm 0.55$ | $36.39 \pm 0.65$ | $\mathbf{23.15 \pm 0.15}$ | $\mathbf{16.84 \pm 0.08}$ |

Table 10: Performance comparison on the CIFAR100 Exp-10 dataset.

| Metric | Standard Accuracy | | Robust Accuracy | |
|---|---|---|---|---|
| Method | Overall | Under | Overall | Under |
| CE | $41.88 \pm 0.36$ | $31.30 \pm 0.57$ | $16.62 \pm 0.03$ | $11.22 \pm 0.21$ |
| Focal | $41.64 \pm 0.51$ | $31.02 \pm 0.71$ | $16.29 \pm 0.18$ | $10.97 \pm 0.34$ |
| LDAM | $41.55 \pm 0.60$ | $31.74 \pm 0.91$ | $20.20 \pm 0.20$ | $14.71 \pm 0.51$ |
| DRCB-CE | $43.89 \pm 0.26$ | $37.28 \pm 0.29$ | $16.90 \pm 0.19$ | $13.62 \pm 0.14$ |
| DRCB-Focal | $43.38 \pm 0.30$ | $36.17 \pm 0.57$ | $16.04 \pm 0.18$ | $12.56 \pm 0.27$ |
| DRCB-LDAM | $43.36 \pm 0.48$ | $39.27 \pm 0.72$ | $20.36 \pm 0.30$ | $17.63 \pm 0.38$ |
| SRAT-CE | $\underline{45.84 \pm 0.18}$ | $41.72 \pm 0.53$ | $21.20 \pm 0.15$ | $\mathbf{19.23 \pm 0.36}$ |
| SRAT-Focal | $\mathbf{46.38 \pm 0.28}$ | $\mathbf{42.53 \pm 0.79}$ | $\underline{20.09 \pm 0.25}$ | $17.83 \pm 0.56$ |
| SRAT-LDAM | $44.98 \pm 0.33$ | $40.39 \pm 0.69$ | $\mathbf{21.83 \pm 0.33}$ | $\underline{18.99 \pm 0.59}$ |

