# OpenReview forum: "Imbalanced Adversarial Training with Reweighting"
_ICLR.cc/2022/Conference — ICLR 2022 Submitted_

### Official Review · Reviewer_6i9F · 2021-10-25

**Correctness:** 4
**Technical Novelty And Significance:** 3
**Empirical Novelty And Significance:** 4
**Recommendation:** 8
**Confidence:** 4

**Main Review:**

Strengths:

1.	This paper is well written and organized. The proposed method SRAT is well motivated by the empirical observations and theoretical analyses.
2.	The experiments are conducted comprehensively on various imbalanced datasets and the efficacy of SRAT is justified by the improved performance on overall classes and under-represented classes. The visualized feature space clearly demonstrates the influence of feature separation loss.

Weaknesses:
1.	The proposed feature separation loss is somewhat trivial, which is very similar to the objective of self-supervised contrastive learning. There are many other ways to help feature become more separated, e.g., channel-wise activation suppressing [1], large-margin softmax [2]. Therefore, I wonder why the authors here utilize the contrastive loss instead of other methods that help improve feature separability.

[1] Improving Adversarial Robustness via Channel-wise Activation Suppressing, ICLR 2021

[2] Large-Margin Softmax Loss for Convolutional Neural Networks, ICML 2016


**Summary Of The Paper:**

This paper provides two critical observations. The first is that adversarial training has worse performance on under-represented classes than natural training on imbalanced datasets. The second is that conventional reweighting methods that upweight under-represented classes largely hurt the performance on well-represented classes for adversarial training. Further, the authors theoretically analyze the reason for the above observations on a binary-classification case. Motivated by the theoretical analysis, the authors incorporate standard adversarial training with feature separation loss, namely SRAT, which is empirically validated SRAT can improve the performance on imbalanced datasets.

**Summary Of The Review:**

Overall, this paper firstly investigates adversarial training in a new scenario under imbalanced datasets and propose to improve feature separability for adversarial training that should enhance the performance on imbalanced datasets.

---

> ### Author Response · Authors · 2021-11-23
> **Response to Reviewer 6i9F**
>
> Thanks for your positive comments. We address them in detail as follows.
>
> **Q1. There are many other ways to help feature become more separated. Why utilize the contrastive loss instead of other methods that help improve feature separability.**
>
> Answer: The contrastive loss in self-supervised contrastive learning methods can naturally result in a more separable latent feature space via enforcing the learned features of examples from the same class aggregate together in the latent feature space [1]. In this work, our goal is to increase the separability in the learned latent space. Hence, we proposed a feature separation loss inspired by the contrastive loss. Our feature separation loss can be easily implemented and do not increase the computation costs of the framework, and our experimental results verify the effectiveness of our feature separation loss. As one of future direction, we're also very interested in integrating other methods that can increase the feature space separability, such as large-margin softmax, into our SRAT framework.
>
> [1] Chen, Ting, et al. "A simple framework for contrastive learning of visual representations." International conference on machine learning. PMLR, 2020.

---

> > ### Comment · Reviewer_6i9F · 2021-11-29
> > **Post rebuttal**
> >
> > Thank the authors for the detailed response.

---

### Official Review · Reviewer_9erX · 2021-10-27

**Correctness:** 3
**Technical Novelty And Significance:** 2
**Empirical Novelty And Significance:** 3
**Recommendation:** 3
**Confidence:** 4

**Main Review:**

# Strengths:

-  Experimental validation is conducted both in the step and exponential imbalance.
-  The motivation for studying the imbalanced datasets and their adversarial training is clear.
-  The preliminary study in sec. 2 exhibiting the problem is useful.

# Questions:

The following parts were not clear while studying the manuscript:

-  Given that several sections of this work are focused on the reweighting strategy, e.g. sec. 2.2, 4.1, I am wondering about its significance in the first place; is it the most popular in training on imbalanced datasets?
-  The analysis in sec. 3 is on linear models, and I am wondering whether the analysis can be extended to non-linear models (e.g. neural networks used in the experiments) or how it is relevant for the proposed losses in sec. 4. Is this added for intuition only?
-  In addition, the analysis in sec. 3 is not studied for an adversarial training objective (e.g. as in eq. 1), so how is this informative for the losses in sec. 4?
-  What is the contribution of the two losses in isolation? That is, what is the significance of the reweighting scheme alone and what is the contribution of the separation loss alone? There is a small ablation study, in Fig. 6, but I believe this should be extended further and analyzed.
-  The experimental setup seems rather weak for an empirical paper. Namely:
    -  Both datasets only include 10 classes, while in practical scenarios many datasets have orders of magnitude more classes. It is recommended to conduct experiments in CIFAR100 or Tiny-Imagenet.
    -  Both datasets include 32x32 images, while in practice images of higher resolution are used for real-world tasks (that is the stated goal for imbalanced dataset study).
    -  CIFAR10 and SVHN are artificially imbalanced datasets, would it be possible to extend the experiments to naturally imbalanced datasets? It is possible that having a natural imbalance has a semantic meaning that currently is lost with artificial imbalance.
    -  Would the results differ in case the attacks had an L_2 norm instead of an L_infinity? Is it possible to include such experiments in the revised version? Such attacks are also quite popular even in PGD.
    -  What is the computational cost of the separation loss introduced in sec. 4.2? In every iteration it includes a large sum operation in the denominator, which might be costly.
    -  I notice that the LDAM has a marginal improvement in the case of imbalance ratio 10; is there any intuition about this? Does it work better only under extreme imbalance ratio?
    -  Is it possible to conduct multiple experiments and report the mean and variance in the tables and the figures? In my experience, adversarial training can have quite some variance.



# Minor corrections:
-  The ‘well accuracy’ in (almost) all the figures is confusing; for most of the figures it could be unified under a unique legend (e.g. in Fig. 1) that can be placed above the image.
-  ‘The adversarially perturbations’ (sec. 4.1).
-  ‘have been prove superiority’ (sec. 4.1).
-  ‘all input examples excepts’ (sec. 4.2).
-  ‘makes significantly improvement’ (sec. 5.2).
-  ‘using a relative large weights’ (sec. 5.3).


**Summary Of The Paper:**

The paper focuses on adversarial training for imbalanced datasets. Imbalanced datasets emerge frequently in real-world applications, while as the paper demonstrates the adversarial training on the under-represented classes results in weak performance. The paper showcases that when the classes are not separable, then a linear model in under-represented classes is weak. Two modifications in adversarial training are proposed to ameliorate that: a) a weighted average of the loss per sample (depending on the class this sample belongs into), b) an additional loss that aims in maximizing the class separation.

**Summary Of The Review:**

Currently, the paper includes several flaws, but many of them (e.g. the proofreading, or improvement of the legends) can be fixed. If the experimental section is strengthened and the questions clarified, I will consider my rating again. However, I am not convinced yet of the novelty of the proposed model, other than demonstrating that the problem with adversarial training on imbalanced datasets exists.

---

> ### Author Response · Authors · 2021-11-23
> **Response to Reviewer 9erX**
>
> **Q1. Why studying reweighting strategy?**
>
> Answer: As we mentioned in Section 6, reweighting can be regarded as one of two main categories for existing imbalanced learning methods, and several recently proposed reweighting based methods have achieved well-pleasing performance of the imbalanced learning task, such as Focal loss [1] and LDAM loss [2]. These methods can be easily implemented and worked well with deep neural networks. In addition, re-sampling based methods may bring extra challenges for the learning process, like insufficient training data and additional noises. Hence, as the initial effort to investigate the imbalanced adversarial training problem, we start our study from the reweighting strategy and leave other methods as future work.
>
> [1] Lin, Tsung-Yi, et al. "Focal loss for dense object detection." Proceedings of the IEEE international conference on computer vision. 2017.
>
> [2] Cao, Kaidi, et al. "Learning imbalanced datasets with label-distribution-aware margin loss." arXiv preprint arXiv:1906.07413 (2019).
>
> **Q2. The analysis in Sec. 3 is on linear models, whether the analysis can be extended to non-linear models or how it is relevant for the proposed losses in sec. 4.**
>
> Answer: Many model robustness related works also adopted linear models as a simple theoretical model to illustrate different settings in non-linear deep learning models. (see [3][4][5]). Inspired by these work, in Section 3, we utilized linear models to theoretically illustrate that the performance of the reweighting strategy can be affected by the data separability directly. Based on our theoretically results, we concluded that the poor separability makes adversarial training and its reweighted variants cannot achieve satisfactory performance under imbalance scenarios. Hence, in Section 4, we proposed a feature separation loss to make the learned features of examples from the same class well clustered while from different classes well separated.
>
> [3] Tsipras, Dimitris, et al. "Robustness may be at odds with accuracy." arXiv preprint arXiv:1805.12152 (2018).
>
> [4] Schmidt, Ludwig, et al. "Adversarially robust generalization requires more data." arXiv preprint arXiv:1804.11285 (2018).
>
> [5] Ilyas, Andrew, et al. "Adversarial examples are not bugs, they are features." arXiv preprint arXiv:1905.02175 (2019).
>
> **Q3. The analysis in Sec. 3 is not studied for an adversarial training objective (e.g. as in eq. 1), so how is this informative for the losses in sec. 4?**
>
> Answer: In Section 3, we first shown that adversarial training has less ability on separating data than natural training (Figure 3), and then proven that the poor data separability could hurt model's performance under imbalanced scenarios, even with the help of the reweighting strategy. For facilitating the adversarial training under imbalanced scenarios, in Section 4, we proposed a feature separation loss to enforce more separable latent feature space.
>
> **Q4. What is the contribution of the two losses in isolation? That is, what is the significance of the reweighting scheme alone and what is the contribution of the separation loss alone?**
>
> Answer: Experimental results reported in Table 1 to Table 10 have demonstrated the contribution of these two losses separately. The baseline methods "CE", "Focal" and "LDAM" only contain one loss for prediction; baseline methods "CB-Reweight", "CB-Focal" combine the prediction loss with the reweighting strategy; Our proposed methods "SRAT-CE", "SRAT-Focal" and "SRAT-LDAM" include the prediction loss, feature separation loss and reweighting strategy. As shown in these tables, introducing the reweighting strategy is able to improve the models' performance and our proposed methods can improve the performance further.

---

> > ### Author Response · Authors · 2021-11-23
> > **Response to Reviewer 9erX  (Part 2)**
> >
> > **Q5. Questions about Experiments. 1) It is recommended to conduct experiments in CIFAR100 or Tiny-Imagenet; 2) Both datasets include 32x32 images, while in practice images of higher resolution are used for real-world tasks; 3) Would it be possible to extend the experiments to naturally imbalanced dataset; 4) Would the results differ in case the attacks had an $l_2$ norm instead of an $l_{\infty}$? 5) What is the computational cost of the separation loss introduced in sec. 4.2? 6) any intuition about the LDAM has a marginal improvement in the case of imbalance ratio 100; 7) Is it possible to conduct multiple experiments and report the mean and variance in the tables and the figures?**
> >
> > Answer:
> > 1) We have added the performance comparison between our model and baseline methods on two imbalanced CIFAR100 datasets in Appendix A.6.3 of our revised version.
> > 2) The datasets we used in experiments, i.e., CIFAR10, CIFAR100 and SVHN are most commonly used real image datasets in both imbalanced learning and model robustness related research.
> > 3) We create multiple imbalanced datasets from three real image datasets, so the  imbalanced datasets used in our experiments can also be regarded as naturally imbalanced datasets.
> > 4) We verified the effectiveness of our SRAT method under PDG $l_{\infty}$ attack with obvious performance improvement, and we don't think it would be huge performance difference of our model between PGD $l_{\infty}$ attack and $l_2$ attack. We are conducting experiments for evaluating the performance of our model under PGD $l_2$ attack and will update results later.
> > 5) In practice, our feature separation loss would not introduce much extra computation cost as we only calculate it within each mini batch with batch size 128. In addition, we can also apply negative sampling to pick up negative example pairs used in the denominator of Eq. (6), when using a relatively large batch size.
> > 6) LDAM loss also encourages larger margins for under-represented classes to the decision boundary, which has similar goal with our feature separation loss, so it can achieve better performance with the help of the reweighting strategy. As learning from imbalanced datasets with imbalance ratio 100 is much more challenging than from datasets with imbalance ratio 10, comparing with other prediction loss, such as cross entropy loss and Focal loss, the superiority of LDAM loss is more obvious in this case.
> > 7) We updated all result tables to include mean and variance of models’ performance obtained from three times trials with different model initialization in our revised version.
> >
> > **Minor corrections**
> >
> > Answer: Thanks for pointing out these issues and we have corrected them in our revised version.

---

> > ### Comment · Reviewer_9erX · 2021-11-26
> > **Remarks on the responses**
> >
> > I am thankful to the reviewers for their responses.
> >
> > I am wondering what the not 'much computational cost' means precisely. Is the negative sampling applied in the denominator or it can be applied?
> >
> > Regarding the rest of the responses: a) it seems that mainly in extreme imbalance ratio scenario there is a significant improvement (based on the responses), b) the analysis on linear models is purely for intuition, c) $\ell_2$ attacks might have a similar response but no experimental validation is provided.

---

> > > ### Author Response · Authors · 2021-11-27
> > > **Further Response to Reviewer 9erX**
> > >
> > > 1. The negative sampling can be applied in the denominator to speed up the computation process. In our implementation, as the size  of mini batch we adopted is not quite large, i.e., 128, we used all negative pairs in the denominator instead of sampling.
> > >
> > > 2. (a). Yes, the superiority of our method in the extreme imbalance scenario is more obviously. When the imbalance ratio is large, learning useful information from under-represented could be more difficult, as the shortage of data examples. Therefore, the learned representation of examples belonging to the same under-represented class typically cannot cluster well. Based on both our empirically study and theoretically understandings, this kind of poor data separability will greatly effect the performance of the reweighting strategy. On the contrary, our SRAT method can perform much better under the extreme imbalance scenario, due to the existence of our feature separation loss.
> > > (b). The main purpose of analyzing on linear models is to formally explore the relation between data separability and model's performance under imbalanced scenarios. Based on our theoretically understandings, we realized that enforcing more separable feature space could benefit the reweighing strategy on imbalanced datasets, and, hence, we equipped a feature separation loss in our SRAT method.
> > > (c) We compared the performance of two SRAT variants, i.e, SRAT-CE and SRAT-Focal, with their corresponding strongest baseline method using the same prediction loss function, i.e.,  DRCB-CE and DRCB-Focal, under PGD-$l_{2}$ attacks on the CIFAR10 Step-100 dataset. We followed previous work [1] to setup PGD-$l_{2}$ attacks in our experiments and report experimental results below. The results shown here further verify the effectiveness of our SRAT method.
> > >
> > > | method\metric | overall standard acc | under standard acc. | overall robust acc. | under robust acc. |
> > > | :----:|:----: | :----: |:----: |:----: |
> > > | DRCB-CE | $70.78 \pm 1.84$ | $48.57 \pm 4.01$ | $56.00 \pm 2.00$ | $30.39 \pm 4.01$ |
> > > | DRCB-Focal | $71.59 \pm 1.21$ | $50.85 \pm 2.60$ | $57.89 \pm 1.88$ | $34.14 \pm 3.31$ |
> > > | SRAT-CE | $76.27 \pm 1.46$ | $60.76 \pm 3.04$ | $61.83 \pm 1.53$ | $42.78 \pm 2.90$ |
> > >  |SRAT-Focal | $73.73 \pm 0.48$ | $54.68 \pm 1.06$ | $60.12 \pm 0.54$ | $38.01 \pm 1.35$ |
> > >
> > > [1] Maini, Pratyush, Eric Wong, and Zico Kolter. "Adversarial robustness against the union of multiple perturbation models." International Conference on Machine Learning. PMLR, 2020.

---

### Official Review · Reviewer_mB3v · 2021-10-28

**Correctness:** 3
**Technical Novelty And Significance:** 2
**Empirical Novelty And Significance:** 2
**Recommendation:** 3
**Confidence:** 4

**Main Review:**

I find the premise of this paper, and the resulting observations about the effect of dataset imbalance on AT, interesting. My main concerns with the paper are as follows:

(i) Significance of the findings:

- The claims in theorem 3.1 and 3.2 do not seem too surprising to me. These behaviors are clear even when we consider the two extremes: (a) when the two classes are very-well separated there is no difference in their performance (b) when the classes are close by, the majority class will push the decision boundary towards the minority class.
- In light of this, and given that AT can be viewed as enforcing large margin requirements (i.e., decreasing separation between classes), the fact that adversarially trained models are more significantly affected by dataset imbalance follows fairly naturally.

(ii) Motivation for SRAT:  The authors do not provide sufficient justification for the loss term introduced in (6), or explain why the default cross-entropy loss would not already induce clustering to the extent possible.

(iii) Validity of the experimental results:

- The difference between the overall performance of the proposed method and prior approaches is small and inconsistent (e.g., the SVHN results in the Appendix).
- The authors propose using a deferred reweighting training schedule with their method, however it is not clear also apply this to the baseline methods.
- Given that the size of the dataset w.r.t. the minority class is fairly small (50-500) examples, there is likely a great degree of variance between runs (due to different samplings of the dataset). Thus it is not clear whether the improvements are statistically significant. The authors should report confidence intervals across multiple runs to clarify this.
- Theorem 3.2 predicts that for better separated features, the drop in accuracy on the majority class post re-weighting should be less significant. However from Table 1 and 2, it appears that compared to baselines, the majority class suffers more (similar overall performance and greater improvement on the minority class).
- Are all methods trained to convergence? The train (standard/robust) accuracy for the majority/minority classes should be reported in the appendix.

Other comments:

- The authors should clarify what "well" and "under" accuracies refer to in the figure captions.
- The use of t-SNE to justify claims regarding feature separation can be misleading as has been noted in prior work [[https://distill.pub/2016/misread-tsne/](https://distill.pub/2016/misread-tsne/)]. In general, given the high-dimensional nature of the data, conclusions drawn based on their two-dimensional projections could be incorrect. For instance, from Figure 3, it would seem that the two classes are not separable in feature space for adversarially trained models. However, in Figure 2, it is apparent that a linear classifier trained on this feature space does get good standard accuracy.

**Summary Of The Paper:**

This paper studies the effect of adversarial training (AT) in the context of imbalanced datasets. In particular, the authors demonstrate empirically that adversarially trained models tend to have a larger gap in accuracy between well and under represented classes, as compared to their standard counterparts. Moreover, they show that standard reweighting strategies do not considerably alleviate this issue. Motivated by their findings, they propose adding a regularization function to adversarial training to encourage greater feature separation between classes (SRAT), and study its effectiveness on unbalanced datasets.

**Summary Of The Review:**

While the premise of this paper is interesting, I believe that both the motivation for the SRAT method, and its empirical validation could be substantially improved.


## Post-rebuttal Update

I thank the authors for their response to my comments. I have decided to keep my original score as some of my concerns still hold. In particular:
1. With the new results that the authors have added which show the baseline losses (CE, LDAM, focal) with deferred reweighting, the difference between the proposed approaches and prior work is even less significant and lies entirely within confidence intervals.
2. Figure 15 does not address my concern about the over/under accuracies in Table 1 and 2. In particular, if SRAT indeed led to better separation, one would expect a comparable or lower drop in over accuracy compared to prior work. However, this seems not to be the case as per Table 1 and 2. For example, DRCB-focal and SRAT-focal, have similar overall accuracy, but the latter has higher under (and thus lower over) accuracy. More broadly, this links to my other concern that the loss proposed is not sufficiently motivated, and it is unclear that it leads to better clustering of features.

---

> ### Author Response · Authors · 2021-11-23
> **Response to Reviewer mB3v**
>
> Thanks for your valuable comments. We address them in detail as follows.
>
> **Q1. Significance of the findings.**
>
> Answer: In our preliminary study (Section 2), we observed that naturally trained and adversarially trained models show different behaviors under imbalanced scenarios. We hypothesized the different ability of naturally trained and adversarially trained models on separating the data could be one possible reason causing this phenomenon. Hence, in Section 3, we aimed to theoretically verify this hypothesis. As far as we know, we are the first work to theoretically prove this phenomenon. By formally analyzing the properties of linear models under diverse imbalanced scenarios, we got an conclusion that the poor separability can make adversarial training and its reweighted variants extremely difficult to achieve good performance under imbalance data distribution. Based on both empirically results and theoretically findings, we proposed our SRAT framework and verified the effectiveness of SRTA via comprehensive experiments.
>
> **Q2. Lacking sufficient justification for the loss term introduced in (6), or explain why the default cross-entropy loss would not already induce clustering to the extent possible.**
>
> Answer: Several recent imbalanced learning work proposed to utilize new designed loss function, such as LDAM loss [1], to replace the default cross entropy loss to obtain larger margins for under-represented classes, as the cross-entropy loss cannot induce good clusterings for those under-represented classes under imbalanced scenarios, In addition, in our preliminary study, we found that, comparing with natural training, the adversarial training produces presents obviously poorer separability on the learned features. Therefore, based on the aforementioned two reasons, we introduced a feature separation loss (Eq. (6)) into our final objective function (Eq. (7)) to enforce the learned feature space can be as separable as possible. As shown in our experimental results, comparing with baselines methods utilizing cross entropy loss, such as "CE", "CB-Reweight" and "DRCB-CE", our model "SRAT-CE" achieved improved performance, which verified the effectiveness of our feature separation loss.
>
> [1] Cao, Kaidi, et al. "Learning imbalanced datasets with label-distribution-aware margin loss." arXiv preprint arXiv:1906.07413 (2019).
>
> **Q3. Validity of the experimental results. 1) The difference between the overall performance of the proposed method and prior approaches is small and inconsistent; 2) It is not clear whether the deferred reweighting training schedule is also applied to the baseline methods; 3) The authors should report confidence intervals across multiple runs to clarify the improvements; 4) Theorem 3.2 predicts that for better separated features, the drop in accuracy on the majority class post re-weighting should be less significant. However from Table 1 and 2, it appears that compared to baselines, the majority class suffers more. 5) Are all methods trained to convergence?**
>
> Answer:
> 1) Although our model cannot outperform all baseline methods under all imbalanced settings on SVHN based imbalanced datasets with marginal improvement, in most settings, our model variants achieved better performance, comparing with baseline methods which utilize the same prediction loss. In addition, in our revised version, we added performance comparisons between our model and baseline methods on imbalanced CIFAR100 datasets. The experimental results we obtained from imbalanced CIFAR100 datasets verify the effectiveness of our SRAT method further.
> 2) In our experiments, we also applied the deferred reweighting training schedule to some baseline methods with different loss functions and denoted them as "DRCB-CE", "DRCB-Focal" and "DRCB-LDAM", respectively.
> 3) We reported the mean and variance of models' prediction performance obtained from multiple trials with different model initialization in our revised version.
> 4) In order to investigate the impact of the weight values, we reported model's performance on well-represented classes and under-represented classes in Figure 15 of our revised version. As shown in this Figure, the increasing of the weights for under-represented classes, the model’s performance on under-represented classes can be increased greatly, with a relatively small performance drop on well-represented classes. Based on these experimental results, we concluded that our SRAT method indeed facilitate the reweighting strategyin adversarial training under imbalanced scenarios.
> 5) Yes. Based on the ResNet18 architecture, we set 200 training epochs to train all methods with initial learning rate 0.1 and apply two times learning rate decay at epoch 160 and 180 with the ratio 0.01 to make sure the trained can be trained converge.

---

### Official Review · Reviewer_2Vzh · 2021-11-03

**Correctness:** 3
**Technical Novelty And Significance:** 2
**Empirical Novelty And Significance:** 2
**Recommendation:** 3
**Confidence:** 4

**Main Review:**

Strength: This paper provide a novel method SRAT for adversarial training on imbalanced dataset, SRAT outperformed the vanilla adversarial training method on imbalanced CIFAR10 benchmark. The authors provide enough details and code for reproducing the results.

Weakness:  The experiments in this paper is poorly conducted.
a) The authors only evaluate their model on CIFAR10 benchmarks, the effect of the proposed methods on large scale benchmarks e.g. ImageNet should also be considered.
b) Only PGD-10 $l_\infty$ $8/255$ attacks are evaluated in this paper. Different attacks settings e.g. $\epsilon = 4/255$, $l_2$-norm attacks, should also be considered. It would also be better if the authors can evaluate SRAT on other types of attacks listed in [a]. I understand that evaluating SRAT on all types of attacks is computationally expensive, but selecting two or three of the SOTA attacks on image classification will increase the strength of the results.
c) Missing confidence score. The training results of the model is depend on the random initialization and especially, in this paper, the under accuracy is related to the under-represented classes, where the images are sampled randomly from the original class. Thus I believe report the confidence score on under-represented classes with different random seed is necessary.

I have a question regarding to the proof of Lemma 1, the authors omit the detailed calculation of comparison of Eq. (8) and Eq. (9), but from my perspective, the errors of Eq. (9) are less than Eq. (8) is not so obviously. Although I agree that intuitively $w=\mathbf{1}$ should be the optimal classifier, the detailed discussion should be added for better understanding the proof. Maybe assuming $\mu = (\eta,0,…,0)$ makes it easier.

I also don’t understand the proof of theorem 3.2, when calculating the $Error_{test}(f^*)$, the authors canceled $Pr.(y=-1)$ and $Pr(y=+1)$ in the second line of the formula and optimising a proportion of the original $Error_{test}(f^*)$ (in the third line of the formula). However, as $Pr.(y=+1) = KPr.(y=-1)$, there is no way to cancel them together without introducing a constant K as the coefficient of $Pr.(N(0,1)<(b-d\eta)/(d\sigma))$. I think the optimising objective should be $Pr.(N(0,1)<-(b+d\eta)/(d\sigma))+KPr.(N(0,1)<(b-d\eta)/(d\sigma))$ instead of $Pr.(N(0,1)<-(b+d\eta)/(d\sigma))+Pr.(N(0,1)<(b-d\eta)/(d\sigma))$.

Besides, theorem 3.1 and theorem 3.2 hold with only a large enough imbalance ratio $K$, which is grown exponentially with $\eta^2/(\sigma_1\sigma_2)$. I’m worried about the scale of $K$ in empirical dataset e.g. CIFAR10, due to the curse of dimension in high dimensional space, $\sigma_1$ and $\sigma_2$ has to be small enough ($O(1/\sqrt{d})$) in order to locate in the input space $[0,1]^{d}$, in this case $K$ would become $O(e^d)$, which is invalid comparing to the scale of CIFAR10 dataset. Thus the empirical algorithm of SRAT is not theoretically guaranteed.

[a] Xu et al Adversarial Attacks and Defenses in Images, Graphs and Text: A Review


**Summary Of The Paper:**

This paper propose a new adversarial training algorithm, Separable Reweighted Adversarial Training (SRAT), for imbalanced datasets.

**Summary Of The Review:**

Please check my review details. I would rate the paper as: score 4 weakly rejected

I will consider to improve the score if the authors can address my concerns.

---

> ### Author Response · Authors · 2021-11-23
> **Response to Reviewer 2Vzh**
>
> Thanks for your valuable feedback. We address them in detail as follows.
>
> **Q1. Proposed model is only evaluated on one dataset, CIFAR10.**
>
> Answer: Due to the limited space, we only provided part of experimental results on imbalanced CIFAR10 dataset in the main text. However, besides CIFAR10 dataset, we also evaluated our proposed model with representative baseline methods on multiple imbalanced datasets created from CIFAR10 and SVHN datasets and reported related results in Appendix A.6. In our revised version, we conducted performance comparison on two imbalanced CIFAR100 datasets and added new experiment results in Appendix A.6.3. Comparing with baseline methods, our SRAT method improved the trained model's performance on two imbalanced CIFAR100 datasets clearly, especially for the SRAT-CE and SRAT-Focal variant. Hence, based on all experimental results, we conclude that our SRAT method is able to facilitate the reweighting strategy in adversarial training under imbalanced scenarios.
>
> **Q2. Only PGD-10 $l_{\infty}$ attacks are evaluated.**
>
> Answer: As the PGD attack has been proven to be the strongest first-order attack, and evaluating a defense model on all types of attacks is computationally expensive, we adopted the most common settings of the PGD-$l_{\infty}$ attack used in many related works [1][2] on CIFAR10, CIFAR100 and SVHN datasets to evaluate the performance of our method. To evaluate our SRAT method more comprehensively, we also compared the performance of two SRAT variants, i.e, SRAT-CE and SRAT-Focal, with their corresponding strongest baseline method using the same prediction loss function, i.e.,  DRCB-CE and DRCB-Focal, under PGD-$l_{2}$ attacks on the CIFAR10 Step-100 dataset. We followed previous work [3] to setup PGD-$l_{2}$ attacks in our experiments and report experimental results below. The results shown here further verify the effectiveness of our SRAT method.
>
> | method\metric | overall standard acc | under standard acc. | overall robust acc. | under robust acc. |
> | :----:|:----: | :----: |:----: |:----: |
> | DRCB-CE | $70.78 \pm 1.84$ | $48.57 \pm 4.01$ | $56.00 \pm 2.00$ | $30.39 \pm 4.01$ |
> | DRCB-Focal | $71.59 \pm 1.21$ | $50.85 \pm 2.60$ | $57.89 \pm 1.88$ | $34.14 \pm 3.31$ |
> | SRAT-CE | $76.27 \pm 1.46$ | $60.76 \pm 3.04$ | $61.83 \pm 1.53$ | $42.78 \pm 2.90$ |
>  |SRAT-Focal | $73.73 \pm 0.48$ | $54.68 \pm 1.06$ | $60.12 \pm 0.54$ | $38.01 \pm 1.35$ |
>
> [1] Zhang, Hongyang, et al. "Theoretically principled trade-off between robustness and accuracy." International Conference on Machine Learning. PMLR, 2019.
>
> [2] Wu, Dongxian, Shu-Tao Xia, and Yisen Wang. "Adversarial weight perturbation helps robust generalization." arXiv preprint arXiv:2004.05884 (2020).
>
> [3] Maini, Pratyush, Eric Wong, and Zico Kolter. "Adversarial robustness against the union of multiple perturbation models." International Conference on Machine Learning. PMLR, 2020.
>
> **Q3. Missing confidence score.**
>
> Answer: In our revised version, We updated all result tables to include mean and variance of models' performance obtained from three times trials with different model initialization. Based on our experimental results, we observed that our SRAT method outperforms all baseline methods under most imbalanced scenarios.
>
> **Q4. Missing detailed calculation of comparison of Eq. (8) and Eq. (9) in the proof of Lemma 1.**
>
> Answer: We have provided more details of proofing Lemma 1 in Appendix 3.1 of our revised version.
>
> **Q5. Why terms $Pr(y = -1)$ and $Pr(y = 1)$ are canceled directly in the proof of theorem 3.2?**
>
> Answer: In Theorem 3.2, we aimed to prove that the poor data separability will cause the performance drop of the well-represented class on the test data set, when upweighting the weight of under-represented class. Hence, when calculating the prediction error of the model trained by the reweight value $\rho$ on an uniformly distributed test data set, terms $Pr(y = -1)$ and $Pr(y = 1)$ can be canceled directly as $Pr(y = -1) = Pr(y = 1)$.
>
> **Q6. Theorem 3.1 and Theorem 3.2 hold with only a large enough imbalance ratio.**
>
> Answer: Theorem 3.1 and Theorem 3.2 proven that the poor separability can be one important reason that makes adversarial training and its reweighted variants extremely difficult to achieve good performance under imbalance data distribution, with the assumption that the imbalance ratio $K$ is large enough. To investigate whether our SRAT method, which is inspired from our theoretically analysis, can be effective on imbalanced datasets with small imbalance ratios, in our revised version, we tested our SRAT method on a series of imbalanced CIFAR10 datasets created by setting the value of the imbalance ratio $K$ from 5 to 100. As shown in Figure 16, comparing with DRCB-Focal method, our SRAT-Focal variant can present consistent benefits under different imbalanced scenarios, even when the imbalance ratio $K$ is small.

---

### Decision · Program_Chairs · 2022-01-20

**Decision:**

Reject

**Comment:**

Three out of the four reviewers raised various concerns on motivation clarify, result significance, and unclear writing. While the authors provided their rebuttals, unfortunately no reviewer seems to have changed their mind. AC reads the paper and agreed this paper perhaps needs major revision before publishing in a major venue. However, the technical ideas are still interesting and promising; the authors are suggested to carefully take into account reviewer comments during revision.